# Heterogeneity of human bone marrow and blood natural killer cells defined by single-cell transcriptome

Chao Yang[1,2], Jason R. Siebert[1,2], Robert Burns[3], Zachary J. Gerbec[1,2], Benedetta Bonacci[4], Amy Rymaszewski [5], Mary Rau[6], Matthew J. Riese[2,7,8], Sridhar Rao [5,9,10], Karen-Sue Carlson[8,11], John M. Routes[5], James W. Verbsky[5], Monica S. Thakar[1,5] & Subramaniam Malarkannan [1,2,5,8]

Natural killer (NK) cells are critical to both innate and adaptive immunity. However, the development and heterogeneity of human NK cells are yet to be fully defined. Using single-cell RNA-sequencing technology, here we identify distinct NK populations in human bone marrow and blood, including one population expressing higher levels of immediate early genes indicative of a homeostatic activation. Functionally matured NK cells with high expression of *CX3CR1*, *HAVCR2* (TIM-3), and *ZEB2* represents terminally differentiated status with the unique transcriptional profile. Transcriptomic and pseudotime analyses identify a transitional population between CD56$^{bright}$ and CD56$^{dim}$ NK cells. Finally, a donor with GATA2$^{T354M}$ mutation exhibits reduced percentage of CD56$^{bright}$ NK cells with altered transcriptome and elevated cell death. These data expand our understanding of the heterogeneity and development of human NK cells.

[1] Laboratory of Molecular Immunology and Immunotherapy, Blood Research Institute, Versiti, Milwaukee, WI, USA. [2] Departments of Microbiology and Immunology, Medical College of Wisconsin, Milwaukee, WI, USA. [3] Bioinfomatics Core, Blood Research Institute, Versiti, Milwaukee, WI, USA. [4] Flow Cytometry Core, Blood Research Institute, Versiti, Milwaukee, WI, USA. [5] Departments of Pediatrics, Medical College of Wisconsin, Milwaukee, WI, USA. [6] Departments of Surgery, Medical College of Wisconsin, Milwaukee, WI, USA. [7] Laboratory of Lymphocyte Biology, Blood Research Institute, Versiti, Milwaukee, WI, USA. [8] Departments of Medicine, Medical College of Wisconsin, Milwaukee, WI, USA. [9] Laboratory of Stem Cell Transcriptional Regulation, Blood Research Institute, Versiti, Milwaukee, WI, USA. [10] Departments of Cell Biology, Neurobiology, and Anatomy, Medical College of Wisconsin, Milwaukee, WI, USA. [11] Laboratory of Coagulation Biology, Blood Research Institute, Versiti, Milwaukee, WI, USA. Correspondence and requests for materials should be addressed to S.M. (email: subra.malar@bcw.edu)

Natural killer (NK) cells are type 1 innate lymphoid cells (ILCs) that eliminate virally infected or transformed cells by direct cytotoxicity and cytokine production[1,2]. Loss of NK cells renders patients prone to viral infection, especially herpes viruses[3]. NK cells have proven to be safe and effective mediators of cellular immunotherapy[4,5]. Despite their clinical relevance, the developmental progression, transcriptomic landscape, and heterogeneity are far from fully defined.

Unlike murine NK cells, which primarily develop in the bone marrow (BM), human NK cells are shown to differentiate in the secondary lymphoidal organs where most of their precursors and immature cells are located[6,7]. Recently, the definition of the NK cell-restricted progenitor population has been redefined as Lin$^-$CD34$^+$CD38$^+$CD123$^-$CD45RA$^+$CD7$^+$CD10$^+$CD127$^-$ cells as a consistent fraction of putative NK precursors in secondary lymphoid organs turn out to be ILCs or ILC progenitors[8–14]. Conventionally, majority of the committed NK cells are identified by cell surface expression of CD56 and are further divided into CD56$^{bright}$ and CD56$^{dim}$ NK cells[15,16]. CD56$^{bright}$ NK cells are believed to be the precursors of CD56$^{dim}$ NK cells with a preponderance of evidence supporting this linear progression model[7]. Nevertheless, contradictory reports challenge this dogma[17–19]. Fate mapping experiment using DNA barcode technique in rhesus macaque demonstrates that CD56$^{bright}$ and CD56$^{dim}$ NK cells derive from different precursor populations[18]. Further, individuals with GATA2 heterozygous mutations have been reported to possess only CD56$^{dim}$ NK cells, apparently bypassing the CD56$^{bright}$ stage[17]. CD56$^{bright}$ NK cells have also been proposed to be an independent ILC1 population based on the function and transcriptome similarity between these two populations[19].

Functionally, CD56$^{bright}$ NK cells have an increased capacity of cytokine production compared to CD56$^{dim}$ NK cells, which are potently cytotoxic[6,16]. The CD56$^{dim}$ NK population is further divided into two groups based on the expression of CD57, where CD57$^+$ cells form a terminally mature subset with a greater killing capacity[20,21]. In contrast to this simple CD56- and CD57-based (CD56$^{bright}$ → CD56$^{dim}$CD57$^-$ → CD56$^{dim}$CD57$^+$) developmental paradigm, mass cytometry (CyTOF)-based immune profiling has revealed thousands of phenotypically distinct NK cells depending on the combinatorial expression of 28 cell surface receptors[22]. This contrast emphasizes the importance of further defining the heterogeneity of the NK population using other modalities, including underlying transcriptional divergence.

The recent breakthrough of single-cell RNA-sequencing (scRNA-seq) technology allows us to study the heterogeneity of a given population based on the transcriptome of each cell. In this study, we use droplet-based scRNA-seq technology to explore the development and heterogeneity of human NK cells from BM and peripheral blood. We find a far more significant heterogeneity of human NK cells than previously defined by cell surface markers. The transcriptome-based differentiation analyses support that CD56$^{bright}$ NK cells are the precursors of CD56$^{dim}$ NK cells with identification of a transitional population. Our data provide a transcriptome-based definition of the heterogeneity and development of human NK cells.

## Results

### Single-cell RNA-seq analyses reveal distinct human NK subsets.
To define the heterogeneity, we performed scRNA-seq experiments using NK cells from BM and blood of six and two healthy donors, respectively. Among these, two individuals donated both the BM and blood. Since NK progenitors and some immature NK cells do not have detectable CD56 expression on the cell surface, we sorted Lin$^-$CD7$^+$ cells instead of Lin$^-$CD56$^+$ cells to include

all the developmental stages of NK cells and ILCs[6] (Supplementary Fig. 1). Within the Lin$^-$CD7$^+$ population of BM or blood, about 90% are CD56$^+$ (Supplementary Fig. 2). Importantly, of the remaining CD56$^-$ cells, more than half of them express NKp80 and CD16, indicating that they are mature NK cells that lost CD56 expression on their cell surface (Supplementary Fig. 2)[23]. The remaining CD7$^+$CD56$^-$CD16$^-$NKp80$^-$ cells could be ILCs/NK progenitors, ILCs, immature NK cells, or immature cells with multiple lineages potentials[13].

Initial quality control (QC) revealed high NK cell purity, optimal library assembly, and sequencing. Majority of the sequenced cells had more than 3000 median unique molecular identifiers (UMIs) and a minimum of 1000 genes associated with the cell barcodes (Supplementary Fig. 3A). Most of the cells had <7% of the total gene expression transcribed from mitochondrial genes indicating robust cell viability (Supplementary Fig. 3A). We combined the cells from six BM donors into one group and the two peripheral blood donors into another for analyses. After the QC filtering, we had a total of 5847 BM cells and 3061 blood cells. Initial clustering resulted in nine distinct clusters of Lin$^-$CD7$^+$ cells from BM (Supplementary Fig. 3B). As expected, all the clusters have a similar level of CD7 expression (Supplementary Fig. 3C). Due to the relatively low number of genes profiled per cell from the 10X platform, CD56 (NCAM1) transcripts are not well represented in the dataset (Supplementary Fig. 3C). Therefore, we identified four surrogate markers of CD56 to define NK-lineage cells. These included CD94 (KLRD1) and NKp80 (KLRF1) that are well-defined NK cell markers[6]. We also included NKG7 and GNLY as these are the most differentially expressed genes (DEGs) in NK cells compared to other lineages in the total peripheral blood mononuclear cell scRNA-seq dataset[24]. Overlaying these four markers with our initial clustering revealed that cluster #8 and #9 are not part of the NK cell lineage (Supplementary Fig. 3C). We further demonstrate a high expression of B or dendritic cell-specific markers (IGJ/MZB1/LILRA4) in cluster #8 and T cell-specific markers (CD3D/E/G) in cluster #9 (Supplementary Fig. 3C). Therefore, we excluded cluster #8 and #9 from further analyses to focus on NK-lineage cells.

In the remaining clusters, we had 5567 and 3046 NK cells from the BM and blood, respectively. These NK-lineage cells were grouped into seven BM- and five blood-derived clusters visualized using a t-distributed stochastic neighbor embedding (t-SNE) plot where each dot represents a single cell (Figs. 1a and 2a). Based on the cluster-defining DEGs (Figs. 1b and 2b, Supplementary Data 1 and 2), we named each cluster in the BM samples: "CD56$^{bright}$ NK," "Transitional NK," "Active NK," "Adaptive NK," "Mature NK," "Terminal NK," and "Inflamed NK." (Fig. 1a). The blood samples had similar transcriptional clusters except the "Adaptive NK" and "Inflamed NK" for reasons related to specific donors. The basis of these nomenclatures would be discussed in detail later. As expected, all the clusters have a high expression of NK cell lineage-defining markers (Figs. 1c and 2c). We then calculated the relative proportion of each cluster within the individual donor (Figs. 1d and 2d). The standard deviation among healthy individuals in each cluster was generally smaller than 5% with Cluster #4 and #6 in the BM samples reaching up to 10%. To validate whether all these clusters were universal among the donors, we plotted the percentages of individual clusters within each healthy donor (Figs. 1e and 2e). Most of these clusters were found in all the donor samples apart from the "Inflamed NK" cluster, which was dominated in the 35-year-old male donor sample (Fig. 1e).

### A transitional subset links CD56$^{bright}$ and CD56$^{dim}$ NK cells.
Although we hoped to capture more NK progenitors or

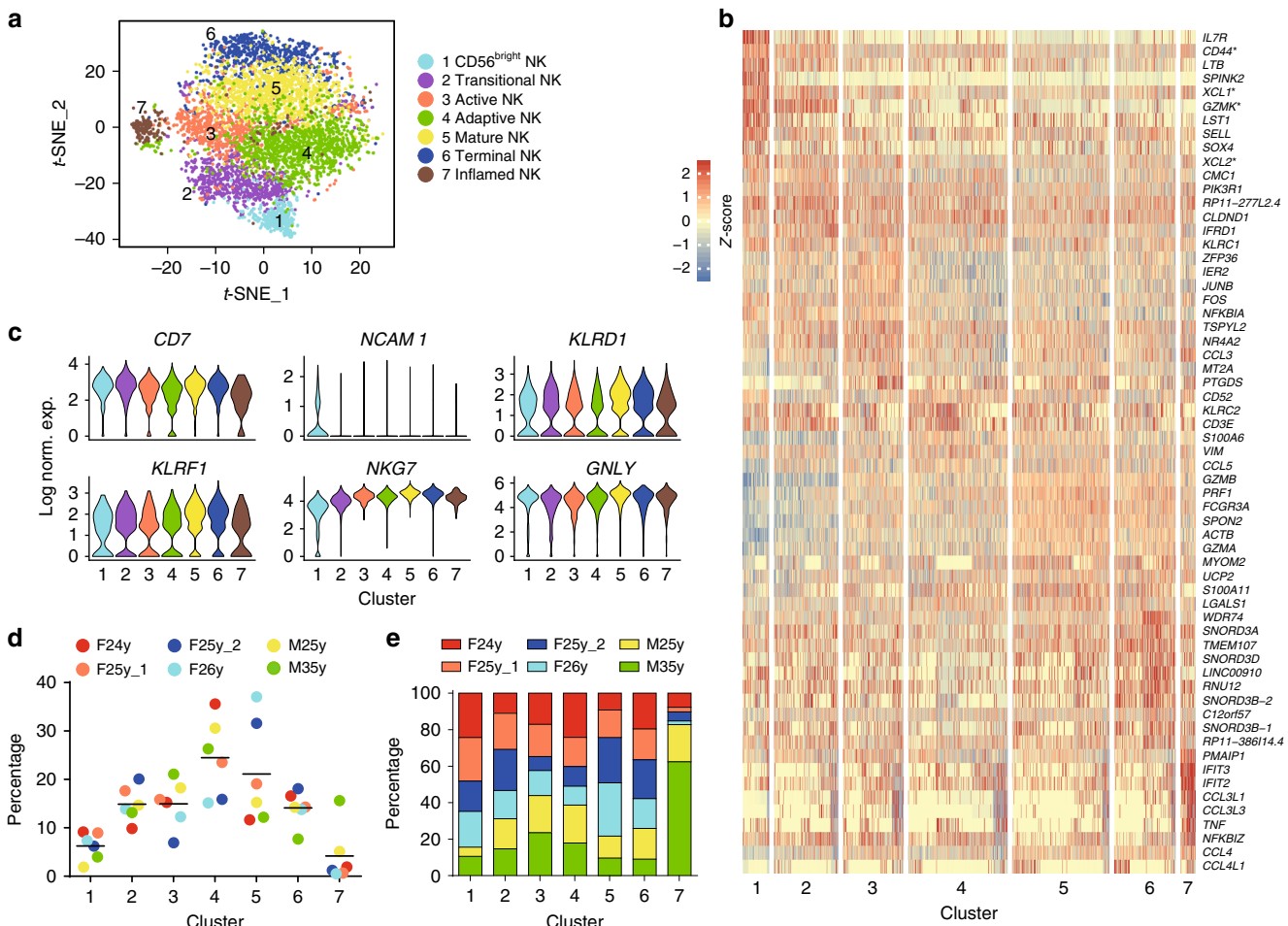

**Fig. 1** Unbiased clustering of human NK cells from BM. **a** Seven distinct human BM NK clusters were numbered, named, and displayed with a *t*-SNE plot. **b** Top 10 up-regulated DEGs (ranked by log fold change) of each cluster were plotted using heatmap. * indicates genes that are DEGs of more than one cluster. **c** Violin plots demonstrate the expression of six NK-lineage-defining markers of each cluster. The *y*-axis represents log-normalized expression value. **d** Composition of the clusters within each donor. **e** Composition of the donors within each cluster. The input cell number from each donor is normalized to be equal. Source data **d**, **e** are provided as a Source Data file. See also Supplementary Figs. 1, 2, 3, and 8

immature NK cells in the BM and blood via sorting CD7+ instead of CD56+ NK cells, we were unable to find an early immature cluster using an unbiased approach. This was presumably due to the low abundance of these populations in the BM and blood[7]. When we plotted the NK-lineage markers overlaying the *t*-SNE cell plot, we did find a small fraction of cells within "CD56bright NK" cluster that had *CD7* expression, but not *KLRD1* (CD94), *KLRF1* (NKp80), *NKG7*, or *GNLY* as compared to the rest of cells (Supplementary Fig. 4A). The analyses of NK cells from secondary lymphoidal organs will give insights into early NK developmental stages.

Most of the molecular profiling studies related to human NK cells have focused on comparing CD56bright with CD56dim NK cells[22,25–28]. We identified CD56bright NK cells in the scRNA-seq data using the high expression of IL7RA (*IL7R*), CD62L (*SELL*), NKG2A (*KLRC1*), and granzyme K (*GZMK*) (Fig. 3a)[29,30]. Further, the minimal expression of CD16A (*FCGR3A*) and CD160 (*CD160*) supports the identity of the "CD56bright NK" cluster (Fig. 3a)[31]. Although killer-cell immunoglobulin-like receptors and CD57 (*B3GAT1*) are only minimally represented in the dataset, as predicted the "CD56bright NK" cluster still had the lowest expression of these mature markers among all the clusters (Supplementary Fig. 4B). The 5 to 10% "CD56bright NK" cluster in the BM and blood matched the percentage of

CD56bright NK cells defined by flow cytometry (Figs. 1d and 2d). We found that the "CD56bright NK" cluster had high *CD44* expression (Fig. 3a). Flow cytometry analyses revealed that all the NK cells from BM and blood are CD44+ (Fig. 3b) and all the CD56bright NK cells are also CD44bright (Fig. 3b). This provides another cell surface maker to differentiate CD56bright NK cells. Moreover, gene-set enrichment analysis (GSEA) revealed that the ribosome gene sets were highly enriched in "CD56bright NK" cluster compared to the rest of the cells (Supplementary Data 3 and 4). Consistent with this result, we found an increased expression of ribosomal subunits in this cluster (Supplementary Data 1 and 2).

*XCL1* and *XCL2* from C chemokine family are also highly expressed in "CD56bright NK" cluster consistent with previous data obtained using mass spectrometry (Fig. 3a)[28]. Recently, NK cell-derived XCL1 has been shown to recruit conventional type 1 DCs to the tumor microenvironment, which is critical to the antitumor immunity[32]. However, activation-induced production of these chemokines has not been explored in human NK cells. Through intracellular staining, we confirmed a higher protein level of XCL1 in CD56bright NK cells compared to the CD56dim NK cells (Fig. 3c). More importantly, when we stimulated freshly isolated NK cells from peripheral blood mononuclear cells with phorbol myristate acetate (PMA) and ionomycin, we found a

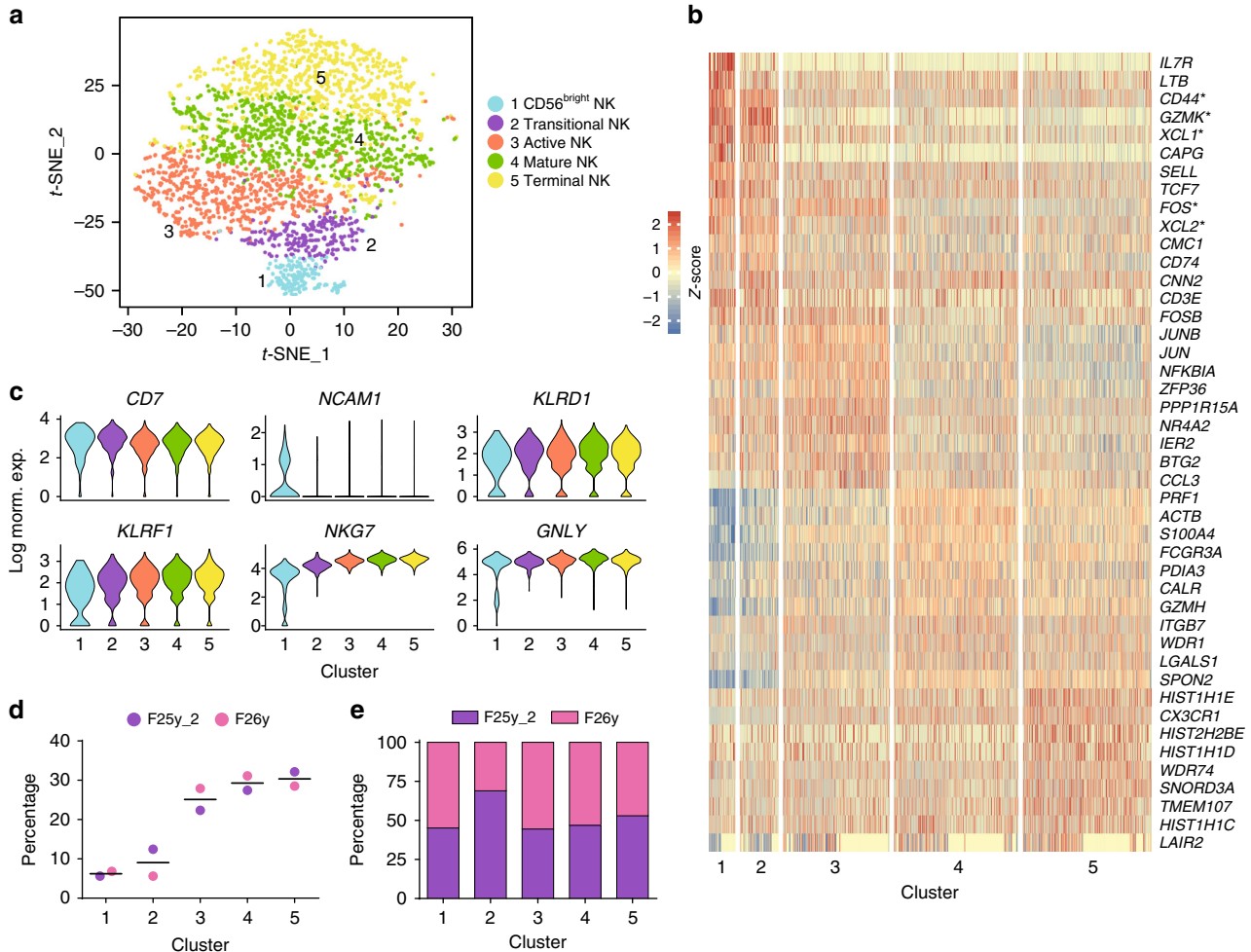

**Fig. 2** Unbiased clustering of human NK cells from the blood. **a** Five distinct human blood NK clusters were numbered, named, and displayed with a *t*-SNE plot. **b** Top 10 up-regulated DEGs (ranked by log fold change) of each cluster were plotted using heatmap. * indicates genes that are DEGs of more than one cluster. **c** Violin plots demonstrate the expression of six NK-lineage-defining markers of each cluster. The *y*-axis represents log-normalized expression value. **d** Composition of the clusters within each donor. **e** Composition of the donors within each cluster. The input cell number from each donor is normalized to be equal. Source data for **d**, **e** are provided as a Source Data file. See also Supplementary Figs. 1, 2, 3, and 8

robust induction of XCL1 expression in both CD56[bright] and CD56[dim] NK cells (Fig. 3c). Again, we found higher production of XCL1 in the CD56[bright] compartment compared to the CD56[dim] NK cells (Fig. 3c). On the contrary, interleukin-12 (IL-12)- and IL-18-mediated stimulation only induced the production of interferon-γ (IFN-γ) but not XCL1 (Fig. 3c and Supplementary Fig. 4C). Given the prevalence of CD56[bright] NK cells in the secondary lymphoid organs, XCL1 and XCL2 production from these NK cells may be necessary in clearing infections and antitumor immunity.

Through the molecular definition of the "CD56[bright] NK" cluster, we found another cluster that is transcriptionally similar to the "CD56[bright] NK" cluster as indicated by the small Euclidean distance between these two clusters (Fig. 3d and Supplementary Fig. 4D). We examined the DEGs and found that this cluster down-regulated several hallmark genes of CD56[bright] NK cells and up-regulated genes associated with CD56[dim] NK cells, in particular, CD16A (*FCGR3A*) (Fig. 3a). This strongly implied the existence of a transitional stage between CD56[bright] and CD56[dim] NK cells. Therefore, we named this subset as the "Transitional NK" cluster. To further explore, we calculated the module scores based on the up- or down-regulated genes associated with the "CD56[bright] NK" cluster or the "Mature NK" cluster, transcriptional representative of CD56[bright] and functionally mature CD57[+]

NK cells, respectively. Module score calculates the difference in the expression between a gene set of interest and another random selection of genes from the transcriptome, and indicates whether gene set of interest is expressed at a higher or lower level than the average[33]. For both up- and down-regulated genes, the "Transitional NK" cluster had an intermediate level of featured gene expression related to CD56[bright] or CD57[+] NK cells (Fig. 3e and Supplementary Fig. 4E). With the availability of the bulk RNA-seq dataset of the three established NK subsets[34], we decided to utilize the DEGs from CD56[bright] and CD56[dim]CD57[+] NK cells and evaluate the expression level of those genes in all the NK clusters defined in our scRNA-seq dataset through module score function. We found the "Transitional NK" cluster has an intermediate expression of signature genes of either CD56[bright] or CD56[dim]CD57[+] NK cells in both BM and blood samples (Fig. 3f and Supplementary Fig. 4F). Collectively, these data helped to establish the identity of the "Transitional NK" cluster.

**Transcriptome of the active NK subset at steady state**. One of the clusters that we found in the BM is the "Inflamed NK" cluster. This cluster is mainly composed of cells from the 35-year-old male donor who only donated the BM sample (Fig. 1e). The top 10 up-regulated genes associated with this cluster included *IFIT3*,

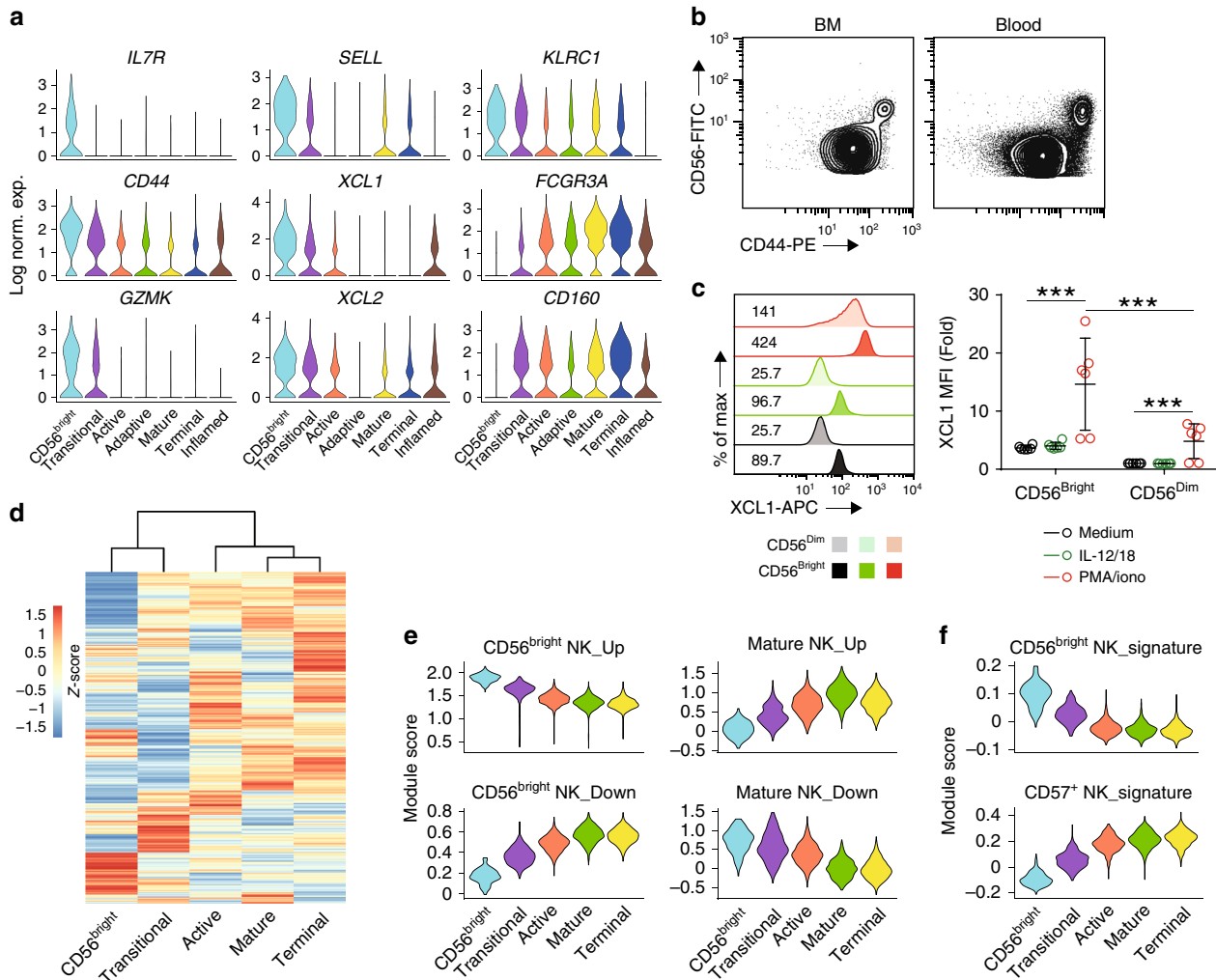

**Fig. 3** Identification of "CD56bright NK" cluster and a potential transitional population between CD56bright and CD56dim NK cells. **a** The expression of genes known to be highly expressed in CD56bright NK cells was displayed using violin plots of the BM samples. The y-axis represents log-normalized expression value. **b** The expression of CD56 and CD44 on NK cells from BM (top) or blood (bottom) of healthy donors were assessed via flow cytometry. Representative flow plot gated on Lin−CD56+ cells were shown. n = 5. **c** The representative histogram plot on the left demonstrates the production of XCL1 from CD56bright and CD56dim NK cells stimulated with medium, IL-12, and IL-18 (10/10 ng/mL), or PMA and ionomycin (50/500 ng/mL) for 6 h. The mean fluorescence intensity (MFI) of each condition was normalized to the medium-stimulated CD56dim NK cells and summarized on the right. n = 6 from three independent experiments. Error bars were shown as standard deviation. Two-way ANOVA was used for the statistical analysis. ***p < 0.001. Source data are provided as a Source Data file. **d** The transcriptome similarity among clusters of the blood sample was evaluated by the Euclidean distance and visualized via heatmap. Each row represents a variable gene among clusters, and each column represents one cluster. **e** Module scores were calculated using up-regulated or down-regulated DEGs of "CD56bright and Mature NK" clusters from blood samples and plotted via violin plots. **f** Bulk RNA-seq-defined DEGs of CD56bright and CD56dimCD57+ NK cells were used to calculate the module score in different clusters of blood samples. See also Supplementary Fig. 4

*IFIT2*, *TNF*, and genes encoding chemokines, indicating stimulation of NK cells presumably by interferons (Fig. 1b). The GSEA results further supported this notion with enrichment of gene sets including IFN-γ signaling, Toll-like receptor (TLR) signaling, and positive regulation of inflammatory responses in the "Inflamed" NK cluster compared to the rest of the cells (Fig. 4a and Supplementary Data 5). This cluster also had a high *CD69* expression (Fig. 4b). All these implied an ongoing active inflammation state when the BM was collected from the 35-year-old male donor. This observation also explained the asymmetric composition of the cluster.

Interestingly, we found a second cluster with cells in activating status that we labeled as the "Active NK." Unlike the "Inflamed NK" cluster, we found the "Active NK" cluster from all the donors in both the BM and blood (Figs. 1e and 2e). The active

nature of cells in this cluster was demonstrated by the fact that the "Inflamed NK" cluster also highly expressed the up-regulated DEGs associated with the "Active NK" cluster (Fig. 4C, left). Conversely, the "Active NK" cluster had a relatively higher module score defined by up-regulated DEGs associated with the "Inflamed NK" cluster (Fig. 4c, right). More than 50% of the DEGs of this cluster was either transcription factors or involved in transcriptional regulation. Importantly, among these genes, we found several that belong to the immediate early genes (IEGs) category including *NR4A2*, *DUSP1*, *FOSB*, *FOS*, *JUN*, and *JUNB* (Fig. 4d and Supplementary Fig. 5A)[35]. These IEGs are induced rapidly in response to stimuli without nascent protein synthesis[36]. GSEA revealed that the "Active NK" cluster compared to the rest of the NK cells were: (1) transcriptionally active; (2) activation of several signaling pathways including KRAS-MAPK, TRAF6-NF-

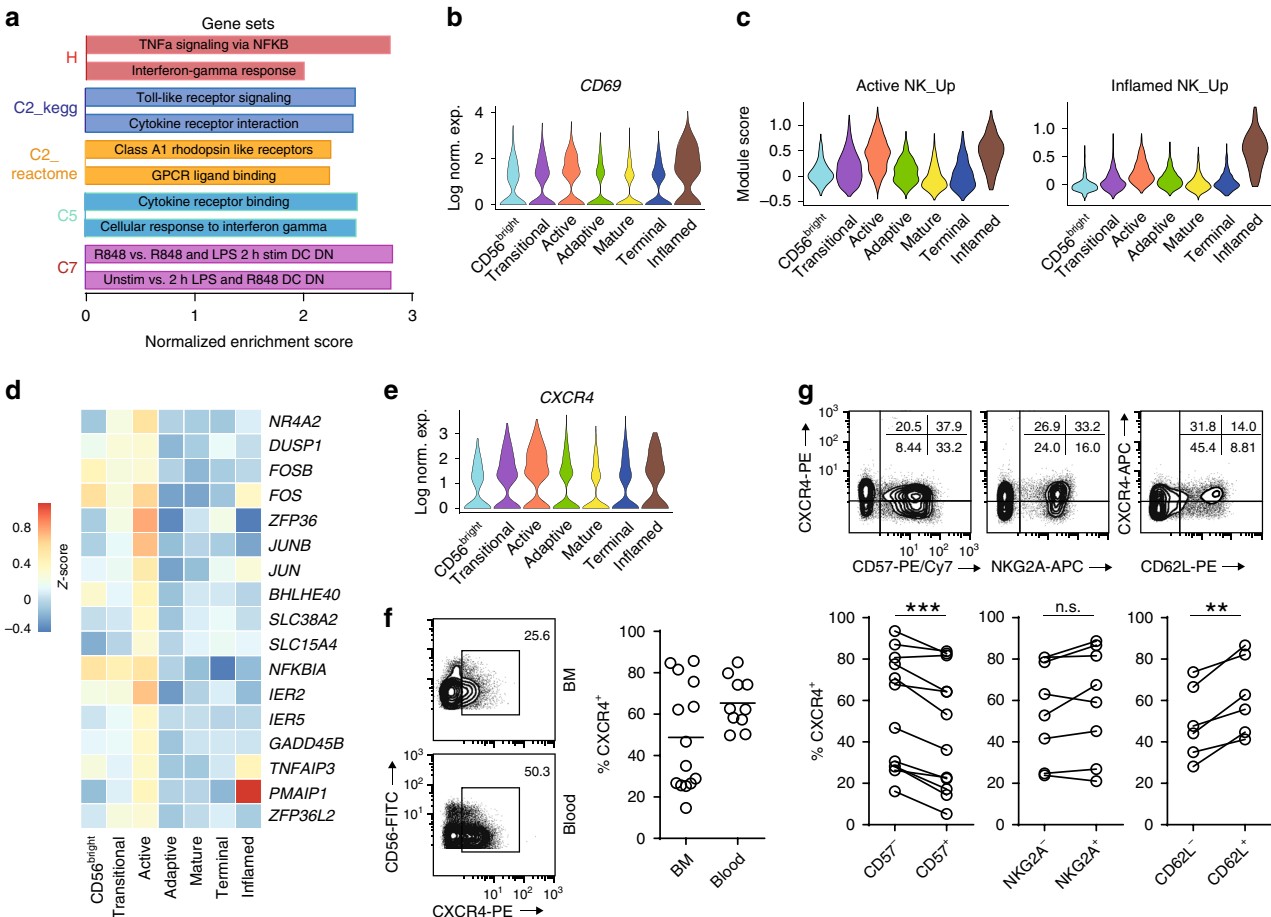

**Fig. 4** Active NK cells with a unique transcriptome profile. **a** Top two enriched gene sets (ranked by normalized enrichment score) of five different datasets from GSEA of the "Inflamed NK" cluster compared to the rest of the cells were plotted. **b** The expression of *CD69* in the BM sample was shown as a violin plot. The y-axis represents log-normalized expression value. **c** Module score was calculated using up-regulated DEGs of "Active NK" (left) or "Inflamed NK" (right) cluster from BM sample and plotted via violin plots. **d** Up-regulated IEGs from "Active NK" cluster were plotted using heatmap of the BM sample. **e** The expression of *CXCR4* in the BM sample was shown as a violin plot. The y-axis represents log-normalized expression value. **f** Percentage of CXCR4+ NK cells (gated on Lin−CD56+ cells) was evaluated via flow cytometry. **g** The expression of CXCR4 in CD57+/−, CD62L+/−, or NKG2A+/− CD56dim NK populations from BM was assessed via flow cytometry (top). Percentage of CXCR4+ cells within each population were quantified (bottom). $n \geq 6$ from two to five independent experiments. Paired Student's t test was used for the statistical analysis. *$P < 0.05$; **$p < 0.01$; ***$p < 0.001$; n.s. stands for "not significant." Source data for **f** and **g** are provided as a Source Data file. See also Supplementary Fig. 5

κB; and (3) activation by various stimuli or receptors including TNF, IL-2, TLR, or GPCR (Supplementary Data 6 and 7). The top 20 hits from the C3 motif gene sets analyses centered around the activation of ATF/CREB (activating transcription factor/cAMP response element-binding protein) and C/EBP (CCAAT enhancer-binding protein) transcription factor families, both of which are known to be involved in the induction of IEGs (Supplementary Data 6 and 7)[36,37]. Collectively, these findings implied an active state of the NK cells in this cluster. We hypothesize that this potentially reflects homeostatic activation of NK cells via certain stimuli, for example, stimulation via trans-presentation of IL-15[38].

Following the identification of this cluster, we sought to identify what cell surface markers define this cluster with flow cytometry. The only cell surface proteins encoded in the DEG set of this cluster were CD69 and CXCR4 (Fig. 4b, e). CD69 is not the optimal cell surface marker to identify the "Active NK" cluster via flow cytometry as we could only detect CD69+ NK cells in the BM (Supplementary Fig. 5B)[39]. On the contrary, we detected CXCR4+ NK cells in both BM and blood via flow cytometry (Fig. 4f). Importantly, we found a higher percentage of CXCR4+

NK cells in the blood than the BM (Fig. 4f), consistent with the scRNA-seq dataset. Based on the DEGs (Figs. 1b and 2b), the Euclidean distance (Fig. 3c, d), and module score analyses (Figs. 3e, f), the "Active NK" cluster is potentially within the CD56dim NK population with a decreased expression of the genes encoding cytolytic proteins compared to the "Mature NK" cluster. Therefore, based on the earlier studies[21,40], we used NKG2A, CD62L, and CD57 to assess the maturity of CD56dimCXCR4+ NK cells in both BM and blood. The cell surface expression of NKG2A and CD62L are known to decrease as NK cells mature, whereas CD57 is generally considered the marker for terminally mature NK cells[30]. Flow cytometry analyses revealed that there are CXCR4+ NK cells in both negative and positive compartments of all three markers; therefore, CXCR4 is not an ideal marker to define developmental stages of NK cells (Fig. 4g and Supplementary Fig. 5C). Nevertheless, we did find a higher percentage of CXCR4+ cells among the CD57− NK cells compared to the CD57+ NK cells in both the BM and blood (Fig. 4g and Supplementary Fig. 5C). This implied that NK cells down-regulate CXCR4 as they become functionally mature, matching the scRNA-seq dataset (Fig. 4e). Consistently, we also

found a higher percentage of CXCR4[+] cells in NKG2A[+] or CD62L[+] NK cells than the negative compartments (Fig. 4g and Supplementary Fig. 5C).

Based on our flow cytometry data, we conclude that the "Active NK" cluster might not represent a unique developmental stage. We further explored this notion by evaluating the expression of IEGs in different subsets (CD56[bright], CD56[dim]CD57[−], CD56[dim]CD57[+]) of human NK cells using the bulk RNA-seq dataset[34]. We found a similar expression level of IEGs among these three subsets of NK cells, which further indicates that "Active NK" cluster may not represent a unique developmental stage (Supplementary Fig. 5D). We reasoned that the "Active NK" cluster might be a mix of cells from different developmental stages receiving stimuli that result in this unique transcriptome profile. The existence of this NK cluster reveals homeostatic activation of NK cells, which may have considerable impacts on NK cell survival, proliferation, or differentiation.

**Unique transcriptomic profile of adaptive NK cells**. Another subset of NK cells that we identified was the "Adaptive NK" cluster, which was found in the BM. This cluster featured with high expression of KLRC2 (NKG2C), which has been used as a marker for the adaptive NK cells resulting from human cytomegalovirus (HCMV) infection (Fig. 1b)[41]. Heatmap of DEGs showed that only a fraction of cells in this cluster had high KLRC2 expression (Fig. 1b). When we plotted KLRC2 expression with individual donors within the "Adaptive NK" cluster, we found that only the 24-year-old female donor had high KLRC2 expression (Fig. 5a). Thus, we conducted the cluster analyses using cells only from the 24-year-old female sample. Due to the low cell number, we only found four clusters, including "CD56[bright] NK," "Active NK," "Mature NK," and "Adaptive NK" (Fig. 5b and Supplementary Fig. 6A). Indeed, the "Adaptive NK" cluster was marked with high KLRC2 expression comprising 40% of the total NK cells from the 24-year-old female sample (Fig. 5c). Previous work from Schlums et al.[42] has discovered that adaptive NK cells from HCMV[+] donors express less FcεRγ (FCER1G), SYK (SYK), EAT-2 (SH2D1B), and PLZF (ZBTB16) compared to other NK subsets. Consistent with this, we found a substantial reduction in the expression of FCER1G and ZBTB16 in the "Adaptive NK" cluster compared to the "Active/Mature NK" clusters from the 24-year-old female sample (Fig. 5d). The expression of SYK and SH2D1B was not well represented in the dataset (Fig. 5d).

Interestingly, although the other five donors did not have cells with high KLRC2 expression (Fig. 5a), some of them still clustered together with the KLRC2[+] adaptive NK cells from the 24-year-old female donor (Fig. 1e). To avoid biased clustering resulting from the adaptive NK cells from the 24-year-old female sample, we removed the cells of the 24-year-old female donor from the "Adaptive NK" cluster and conducted the analyses to test whether this cluster still existed. We were still able to recover all the seven clusters with the original "Adaptive NK" cluster becoming a cluster that only has one up-regulated DEG, CD52 (Supplementary Fig. 6B and Supplementary Data 8). CD52 was the most up-regulated DEG in the original "Adaptive NK" cluster (Fig. 1b and Supplementary Data 1) and had the highest expression in the "Adaptive NK" cluster of the 24-year-old female donor compared to the rest of clusters (Supplementary Fig. 6C). In the absence of NKG2C[+] cells, we still found that this cluster had lower expression of FcεRγ (FCER1G) and PLZF (ZBTB16) compared to the rest of the clusters (Supplementary Fig. 6D). Because of these features, we named this cluster as "Adaptive-like NK" cluster as it was transcriptionally similar to the conventional NKG2C[+] adaptive NK cells. One possible explanation for the

generation of this subset of cells was that they were adaptive NK cells derived from infections other than HCMV.

Next, we focused our analyses on the NK cells from the 24-year-old female donor. Interestingly, we found high CD3E/D/G expression in the "Adaptive NK" cluster (Fig. 5e). We were confident that this cluster was not contaminating T cells based on the 98.8% post-sorting purity of the 24-year-old female sample (Supplementary Fig. 6E). To further confirm the existence of the adaptive NK cells in the 24-year-old female sample, we stained for NKG2C in the cryopreserved samples from four of our six donors. To ensure the exclusion of T cells, we included CD3ε, TCRα/β, and TCRγ/δ in the staining. Evidently, only the 24-year-old female sample had a large percentage of cells expressing NKG2C (Fig. 5f). The percentage of NKG2C[+] NK cells in that sample was about 46%, matching the scRNA-seq data. The increased expression of CD3E/D/G messenger RNA (mRNA) indicated an epigenetic alteration in the adaptive NK cells. Another gene that was highly expressed in the "Adaptive NK" cluster of the 24-year-old female sample was IL32 (Supplementary Fig. 6A). Originally discovered in IL-2-activated human NK cells[43], IL-32 is now appreciated as an important pro-inflammatory cytokine with nine different isoforms through mRNA alternative splicing[44].

To explore whether up-regulation of IL-32 is a unique feature associated with adaptive NK cells, we examined the expression of IL-32 in all the original seven BM NK clusters. Although expressed in all clusters, the "Adaptive NK" and "Inflamed NK" clusters had more IL-32-expressing cells compared to the rest of the clusters (Supplementary Fig. 6F). When we focused on the "Adaptive NK" cluster, we observed high expression of IL-32 not only in the NKG2C[+] NK cells from the 24-year-old female sample but also other donors" cells, indicating that it is not unique to the conventional NKG2C[+] adaptive NK cells (Supplementary Fig. 6G). An important aspect of the adaptive lymphocytes is their rapid and robust functional capacity compared to naive cells. Surprisingly, the "Adaptive NK" cluster has less expression of cytolytic molecules compared to the "Mature NK" cluster (Fig. 1b and Supplementary Fig. 6A). One potential explanation was that the functional molecules were expressed and stored as proteins in these cells, and therefore we no longer detected active transcription of those genes. In line with this idea, GSEA revealed that the "Adaptive NK" cluster was enriched in the endoplasmic reticulum to Golgi transport vesicle gene set compared to the rest of the cells from the 24-year-old female sample (Fig. 5g). Moreover, the positive regulation of IFN-γ production gene set was also enriched in this cluster, indicative of the functional attributes of adaptive NK cells (Fig. 5g).

**Terminally mature NK cells exhibit unique transcriptional profile**. In the current development model, the cell surface expression of CD57 marks the functionally matured human NK cells with the highest cytolytic potential[20,21]. Although CD57 (B3GAT1) is under-represented in our scRNA-seq dataset (Supplementary Fig. 4B), we were able to identify this functionally matured NK population comprising the "Mature and Terminal NK" clusters. The "Mature NK" cluster-featured high expression of molecules important for cytotoxic function (Fig. 6a, Supplementary Data 1 and 2). We found the highest expression of cytolytic molecules in the "Mature NK" cluster, including perforin (PRF1) and granzymes (GZMA, GZMB, GZMH). Optimal cytotoxicity requires cytoskeletal remodeling[45]. Indeed, the expression of β-actin (ACTB), actin-related protein 2/3 complex subunit 2 (ARPC2, assist actin polymerization), and Coronin 1A (CORO1A, remodel F-actin to enable granule release at the synapse), and Cofilin 1 (CFL1, depolymerize and remodel

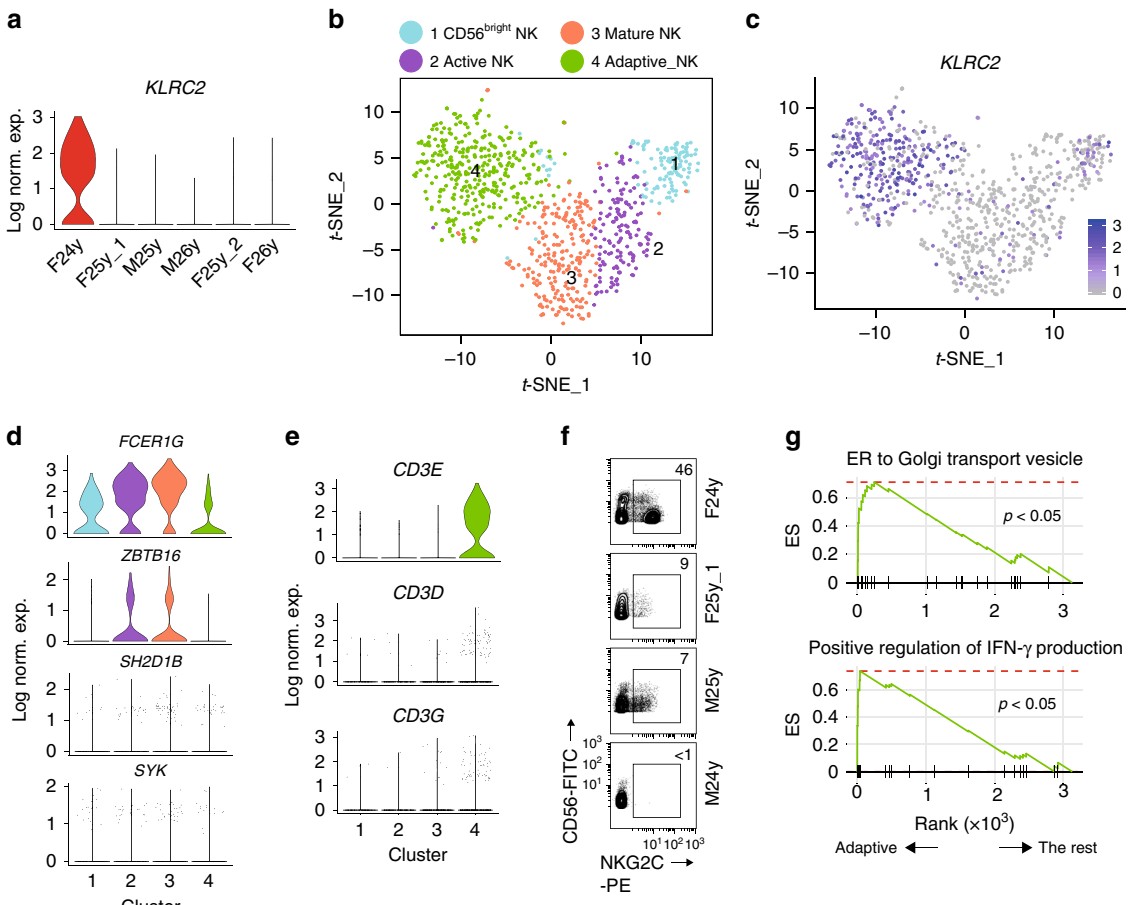

**Fig. 5** Identification of adaptive NK cells from a 24-year-old female donor. **a** The expression of *KLRC2* in each donor cells within the "Adaptive NK" cluster were shown in violin plots. The *y*-axis represents log-normalized expression value. **b** The *t*-SNE plot demonstrates four distinct NK clusters from BM of the 24-year-old female donor. **c** The expression of *KLRC2* of the 24-year-old female sample was overlaid on the *t*-SNE plot. **d** Violin plots demonstrate the expression of *FCER1G*, *ZBTB16*, *SH2D1B*, and *SYK* in four clusters from the 24-year-old female donor sample. The *y*-axis represents log-normalized expression value. **e** Violin plots demonstrate the expression of *CD3E*, *CD3D*, and *CD3G* in four clusters from the 24-year-old female donor sample. The *y*-axis represents log-normalized expression value. **f** Expression of NKG2C on NK cells (gated on PI⁻Lin⁻TCRα/β⁻TCRγ/δ⁻CD56⁺ cells) was assessed by flow cytometry using cryopreserved samples from four donors. **g** Selected gene sets enriched in the "Adaptive NK" cluster compared to the rest of the cells were plotted. See also Supplementary Fig. 6

F-actin) peaked in the "Mature NK" cluster (Fig. 6a)[46–48]. We also found high expression of the negative regulators related to the cytotoxicity such as cystatin F (*CST7*) and profilin 1 (*PFN1*)[49,50]. Also, the cell adhesion and signaling molecules were also up-regulated in this cluster (Fig. 6a). Moreover, GSEA demonstrated enrichment of gene sets, including the apical junction, regulation of actin cytoskeleton, NK cell-mediated cytotoxicity, and structure constituent of the cytoskeleton in the "Mature NK" cluster compared to the rest of the cells (Supplementary Data 9 and 10).

In the "Mature NK" cluster, we found increased expression of genes belongs to the Annexin family (Fig. 6b). The role of this family in human NK cells has not been well characterized. We decided to explore Annexin A1 (*ANXA1*) as this protein has been shown to have an immunosuppressive role in other immune cells and critical for the resolution of inflammation downstream of glucocorticoids[51]. NK cells are known to express Annexin A1[52]. However, the relative protein levels among different NK subsets are unknown. At the protein level, we could detect Annexin A1 intracellularly but not on the cell surface (Fig. 6c and Supplementary Fig. 7A). Consistent with the transcriptome data, we found increased Annexin A1 protein as NK cells mature from CD56^bright → CD56^dimCD57⁻ → CD56^dimCD57⁺ population (Fig. 6c). Next, we explored the glucocorticoids–Annexin A1

axis in the effector functions of human NK cells. Consistent with previous data[53], corticosterone inhibits the generation of IFN-γ from NK cells following activation (Supplementary Fig. 7B). However, Annexin A1 does not affect either PMA and ionomycin- or IL-12- and IL-18-mediated IFN-γ production (Supplementary Fig. 7C). We reasoned that this could due to low expression of Annexin A1 receptor, formyl peptide receptor 2 (*FPR2*) on NK cells compared to the glucocorticoid receptor (*NR3C1*) (Supplementary Fig. 7D). Nevertheless, the high amount of Annexin A1 in the cytosol of human NK cells could potentially regulate the activities of other innate and adaptive immune cells through paracrine activation. To test this possibility, we explored the activation-mediated secretion of Annexin A1 from human NK cells. We found the reduced intracellular level of Annexin A1 in NK cells following stimulation with either IL-12 and IL-18 or PMA and ionomycin indicating potential secretion of this protein (Fig. 6d and Supplementary Fig. 7E). To further test this hypothesis, we utilized enzyme-linked immunosorbent assay (ELISA) to measure the protein level of Annexin A1 in the culture supernatant following stimulation. Indeed, we found an increased level of Annexin A1 in the supernatant (Fig. 6e). These data indicated that in addition to neutrophils, monocytes, and macrophages, NK cells are also a major source of Annexin A1.

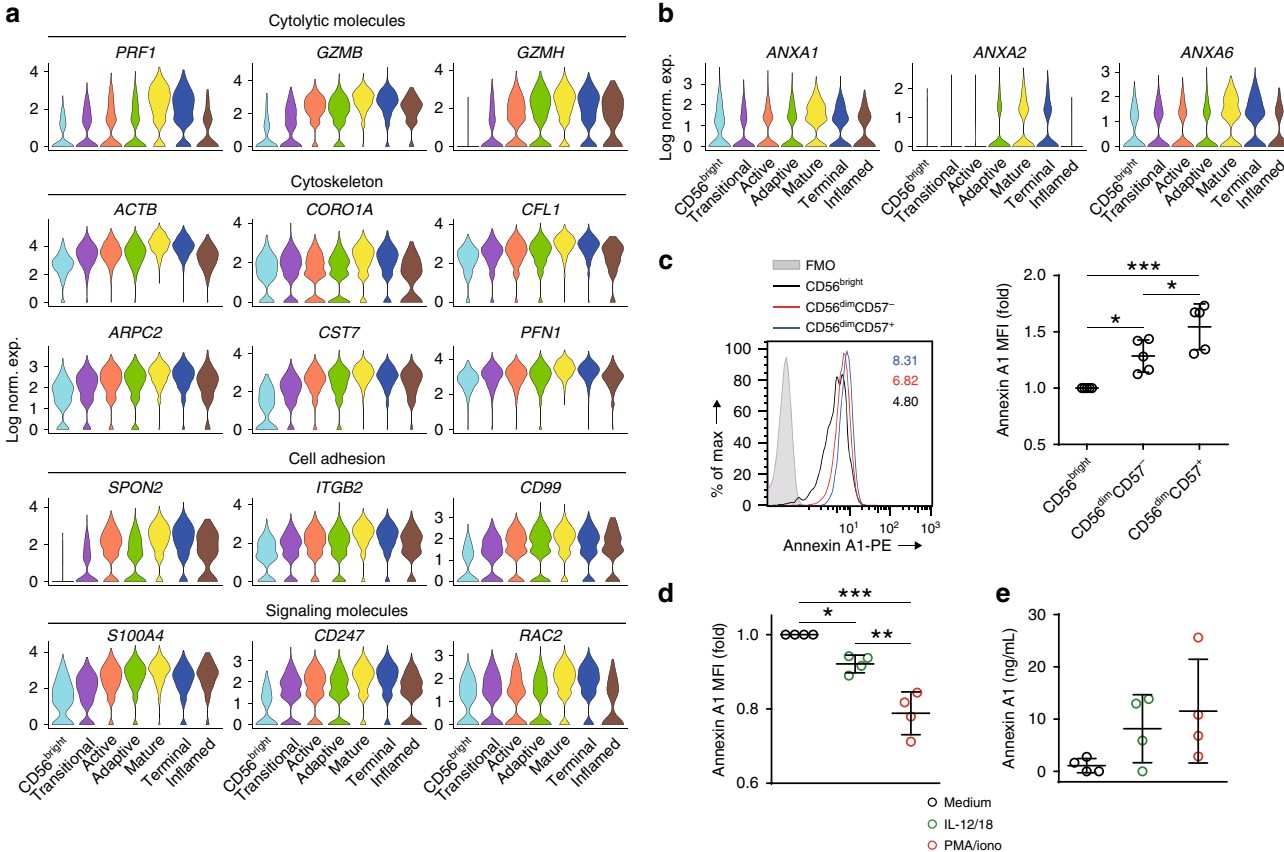

**Fig. 6** Transcriptional signature of functionally mature NK cells. **a** Representative genes associated with functionally mature NK cells were grouped in four different categories and plotted via violin plots of the BM sample. The y-axis represents log-normalized expression value. **b** The expression of *ANXA1*, *ANXA2*, and *ANXA6* in each cluster of the blood sample was shown as violin plots. The y-axis represents log-normalized expression value. **c** The intracellular protein level of Annexin A1 in different NK cell subsets from blood at steady state was assessed by flow cytometry. The representative histogram was shown on the left. FMO stands for fluorescence minus one. The mean fluorescence intensity (MFI) of Annexin A1 in each NK subsets was normalized to CD56$^{bright}$ NK subset and shown on the right. $n = 5$ from three independent experiments. Error bars were shown as standard deviation. One-way ANOVA was used for the statistical analysis. *$P < 0.05$; **$p < 0.01$; ***$p < 0.001$. **d** Isolated NK cells from blood were stimulated with medium, IL-12, and IL-18 (10/10 ng/mL), or PMA and ionomycin (50/500 ng/mL) for 6 h. The protein level of Annexin A1 was assessed following stimulation. The MFI of Annexin A1 in each condition was normalized to the medium-only NK cells. $n = 5$ from three independent experiments. Error bars were shown as standard deviation. One-way ANOVA was used for the statistical analysis. *$P < 0.05$; **$p < 0.01$; ***$p < 0.001$. **e** Isolated NK cells from blood were stimulated with medium, IL-12, and 18 (10/10 ng/mL), or PMA and ionomycin (50/500 ng/mL) for 21 h. Following stimulation, the protein level of Annexin A1 in the supernatant was assessed using ELISA. Source data for **c**, **d**, and **e** are provided as a Source Data file. See also Supplementary Fig. 7

Activation-induced secretion of Annexin A1 from NK cells could influence inflammatory responses.

In both the BM and blood samples, we found another cluster, the "Terminal NK" cluster, has similar expression level of the functional molecules as the "Mature NK" cluster (Figs. 1b, 2b, and 6a) and is transcriptionally similar to the "Mature NK" cluster as indicated by the short Euclidean distance (Fig. 3c, d). The high expression of transcription factor ZEB2 indicated terminal maturity of this cluster (Fig. 7a, b)[54]. To further support the identification of the terminally matured cluster, we demonstrate increased expression of *CX3CR1* and *HAVCR2* (TIM-3), both of which mark the mature NK population with full responsiveness (Fig. 7a, b)[55,56]. There is also a higher percentage of this cluster in the blood compared to the BM (30% vs. 15%). Based on the percentage of the "Mature and Terminal NK" clusters and elevated expression of the functional molecules, we predict these two clusters together form the CD57$^+$ NK cells. Although we found increased expression of *CX3CR1* and *HAVCR2* (TIM-3), the cell surface level of CX3CR1 or TIM-3 is not heterogeneous among the CD56$^{dim}$ NK population (Supplementary Fig. 7F), rendering them insufficient in identifying this "Terminal NK"

cluster by flow cytometry[55,56]. ZEB2 was also not an ideal marker to identify this population as we could detect minimal levels of ZEB2 through intracellular flow analysis (Supplementary Fig. 7G). This is consistent with the fact that the detection of Zeb2 protein in murine model relies on a reporter system[54].

Next, we explored the DEGs between "Mature NK" and "Terminal NK" clusters. Strikingly, we found that the "Terminal NK" cluster expresses distinct members of small nucleolar RNA, C/D box 3 (SNORD3) cluster and various histone subunits at a high level (Fig. 7c, d). GSEA revealed that the ribosome gene set was depleted in the "Terminal NK" cluster compared to the "Mature NK" cluster (Fig. 7e). In contrast, the chromosome maintenance, histone methylation, and telomere maintenance gene set were enriched in the "Terminal NK" cluster (Fig. 7e). We calculated the cell cycle score based on established gene sets[57]. We did not find substantially higher S.Score (module score of genes associated with the S phase of the cell cycle) or G2M.Score (module score of genes associated with the G2M phase of the cell cycle) of the "Terminal NK" cluster indicating this unique transcriptional profile was not due to active cell cycling (Supplementary Fig. 7H). Moreover, GSEA demonstrated

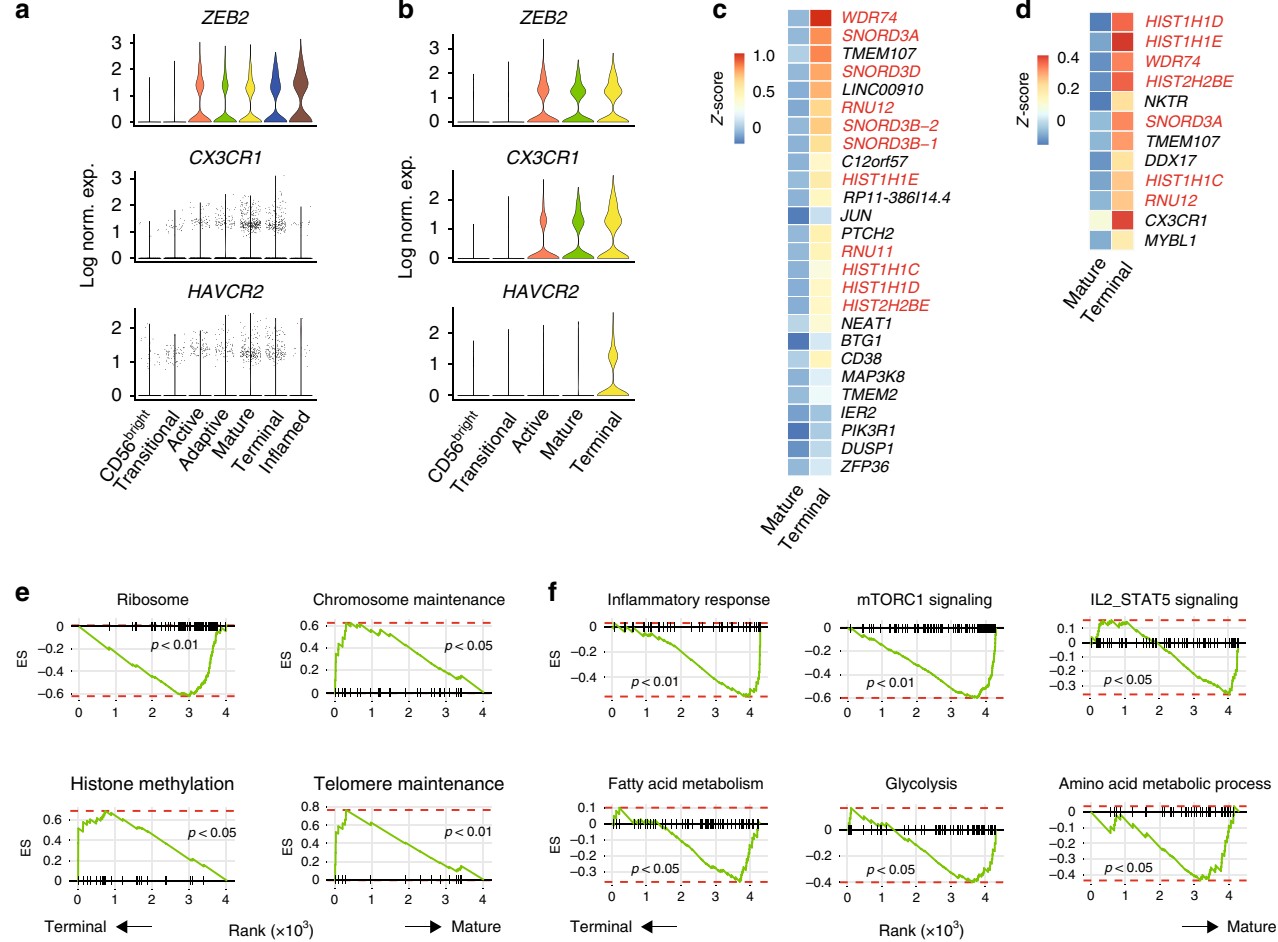

**Fig. 7** Unique transcriptional profile of terminally differentiated NK cells. **a**, **b** The expression of *ZEB2*, *CX3CR1*, and *HAVCR2* in each cluster of the BM (**a**) and blood (**b**) samples were shown as violin plots. The y-axis represents log-normalized expression value. **c**, **d** The expression of all the DEGs between "Mature and Terminal NK" cluster of the BM (**c**) and blood (**d**) samples were plotted via heatmap. Genes related to histones or ribosomes were highlighted in red. **e** Selected gene sets enriched in the "Terminal NK" cluster compared to the "Mature NK" cluster of BM sample. **f** Signaling and metabolic gene sets were reduced in the "Terminal NK" cluster compared to the "Mature NK" cluster of the blood sample. See also Supplementary Fig. 7

decreased inflammatory response, down-regulated mTORC1/STAT5 signaling, and reduced metabolic activity of the "Terminal NK" cluster compared to the "Mature NK" cluster (Fig. 7f). Previously, murine NK cells have been shown to develop into a quiescent state as they become terminally mature. These dampened signaling and metabolic profiles indicated that the "Terminal NK" cluster cells might also be in a quiescent state[58]. To further explore this, we utilized a previously established quiescence gene set using human NK cells and evaluated the expression level of quiescence-associated genes in our BM and blood clusters[59]. Module score indicated no significantly increased expression of these genes in the "Terminal NK" cluster in either BM or blood samples (Supplementary Fig. 7I). The difficulty in identifying this "Terminal NK population by cell surface markers prevented us from exploring it further. Nevertheless, our data indicate heterogeneity of the CD57+ NK cells, which forms the basis for future studies.

**BM and blood contain similar developmental NK clusters.** The presence of similar cluster composition in the BM and blood was consistent with previously reported cell surface marker-defined NK populations[23]. To compare the NK cell clusters between BM and blood, we combined their transcriptional data from the BM and blood of the individual donors (F25y_2 and F26y) and performed a clustering analysis on the combined

populations. Like the previous analyses conducted on the BM or blood separately, we have found "CD56bright NK," "Transitional NK," "Active NK," "Adaptive-like NK," "Mature NK," and "Terminal NK" clusters (Supplementary Fig. 8A, B). Generally, both BM and blood possessed similar heterogeneity of NK cells with the difference in composition (Supplementary Fig. 8C, D). This is in contrast to the organ-specific phenotype between human splenic and blood NK[60]. Consistent with independent analyses shown above, we had more cells in the "Active NK" cluster in blood than BM (Supplementary Fig. 8C, D). Specific to "Adaptive-like NK" cluster, it was mostly composed of cells from BM (Supplementary Fig. 8C, D). This explains why there was no "Adaptive-like NK" cluster in the analyses of blood samples alone. In terms of maturity, the blood possessed a higher percentage of cells from the "Mature NK" cluster than the BM consistent with more CD57+ NK cells in the blood than the BM (Supplementary Fig. 8C, D).

**CD56bright is simulated as the precursor of CD56dim NK cells.** One advantage of scRNA-seq analyses is the ability to simulate the ontology among a developmentally heterogeneous population based on the transcriptional changes during cell differentiation. We used Monocle2 pseudotime trajectory analyses to develop a developmental course of NK cells[61]. As shown in Fig. 8a, b, both BM and blood NK cells demonstrated a relative linear

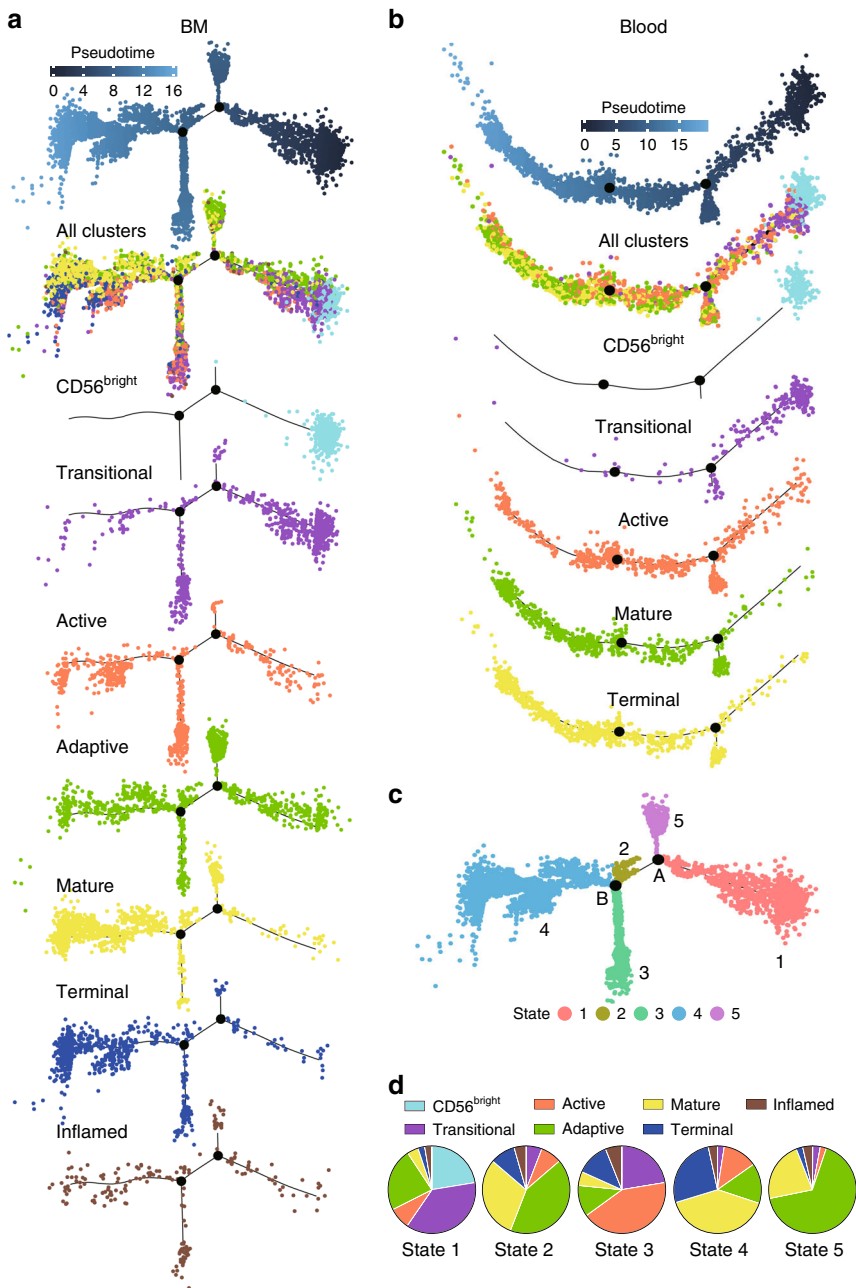

**Fig. 8** Single-cell trajectory analysis reveals a linear NK cell developmental pathway, maturing from CD56[bright] to CD56[dim] NK cells. **a**, **b** Pseudotime trajectory of BM (**a**) or blood (**b**) NK cells with pseudotime, all clusters and individual cluster demonstrated in the trajectory. **c** The BM trajectory was separated into five cell states by branch points A and B. **d** Composition of the clusters within each cell state were demonstrated via pie charts. Source data for are provided as a Source Data file. See also Supplementary Fig. 9

developmental progression with few branches. The "CD56[bright] NK" cluster and "Mature/Terminal NK" cluster dominate the two ends of the progression trajectory in both BM and blood. Based on the current human NK development model, we assigned the "CD56[bright] NK" cluster as the least mature branch in the pseudotime. Importantly, in both the BM and blood samples, cells from the "Transitional NK" cluster develop from the "CD56[bright] NK" cluster and extend in the direction of functionally "Mature NK" cluster. This result not only demonstrates the existence of the transitional subset from CD56[bright] to CD56[dim] NK cells as illustrated above (Fig. 3) but also provides evidence from transcriptional profiling supporting CD56[bright] NK cells as the precursors of CD56[dim] NK cells.

Next, we focused on the analyses of the branches along the developmental trajectory. In the BM sample, branch points A and B separated the cell trajectory into five states (Fig. 8c). We calculated and plotted the percentage contribution of each cluster within each state using the pie charts (Fig. 8d). It was evident that State-1 cells were mainly comprised of "CD56[bright] and Transitional NK" clusters, while the State-4 cells were mainly comprised of "Mature and Terminal NK" clusters. Importantly, we found that State-5 cells branching out from point A was dominated by the "Adaptive NK" cluster (Fig. 8d). Analyzing the genes that were significantly dependent on branch A, we found high expression of *CD3E*, *VIM*, and *CCL5* in State-5 cells (Supplementary Fig. 9A), matching with the gene signatures associated

with "Adaptive NK" cluster (Fig. 1b). Another evident branch was the State-3 cells stemming out from point B. This branch was heavily comprised of "Active NK" cluster (Fig. 8d). The branch point B-dependent genes also revealed high expression of "Active NK" cluster genes in State-3 cells, including *CXCR4*, *JUNB*, *IER2*, *ZFP36*, and *JUN* (Supplementary Fig. 9B). The formation of branches mainly by the "Active and Adaptive NK" clusters demonstrated that the transcriptome footprint of these two clusters differed from the steady-state maturation. When we plotted the composition of each state within each cluster (Supplementary Fig. 9C), evidently, the "CD56$^{bright}$, Transitional, Mature, and Terminal NK" clusters were dominated by one cell state. However, this was not seen in the "Active, Adaptive, or Inflamed NK" clusters, further manifesting the transcriptome alteration induced by the secondary stimuli.

**Altered NK cell transcriptome in a GATA2$^{T354M}$ donor**. Transcription factors are known to play critical roles in lineage specification and cellular differentiation[62]. Among the GATA transcription factor family, it is well established that a haplo-insufficiency in *GATA2* locus results in human NK cell deficiency with specific loss of CD56$^{bright}$ subset[17]. This observation, to some extent, challenges the current developmental dogma that CD56$^{bright}$ NK cells give rise to CD56$^{dim}$ NK cells. However, the failure of flow-based detection of CD56$^{bright}$ NK cells in individuals with *GATA2* mutation does not rule out the possibility of insufficient expression of CD56 on the cell surface, which potentially masks the true CD56$^{bright}$ NK cell identity[29]. To further explore the human NK cell deficiency in individuals with *GATA2* mutation, we used scRNA-seq technology to profile the NK-lineage cells from the blood of a clinically asymptomatic female donor with GATA2$^{T354M}$ mutation. A missense mutation of C → T in a single allele at the 1061 codon of *GATA2* gene results in the conversion of threonine (ACG) to methionine (ATG). T354M transition is the most prevalent *GATA2* mutation identified so far, accounting for about 50% of all the *GATA2* mutation cases[63]. Clinical diagnoses of this donor revealed normal T cell number, moderately decreased absolute B cell number, and the absence of CD56$^{bright}$ NK cells. Considering the clinical diagnoses, we believe this donor is in the early stage of the disease caused by GATA$^{T354M}$ mutation. In our flow analyses, the CD56$^{bright}$ NK cells from this donor were not easily identifiable compared to the healthy control (Fig. 9a). With the help from the CD57 staining, we could only find 2% CD56$^{bright}$ NK cells within the NK population, which is substantially lower than the normal 5–10% range (Fig. 9a). This provided us a unique opportunity to study the human NK cell defect at the early disease stage of *GATA2* mutation.

We performed the scRNA-seq experiment using the Lin$^-$CD7$^+$ cells from the blood of the donor with GATA2$^{T354M}$ mutation. After sequencing, we combined this dataset with the blood samples from two healthy female donors for PCA analyses. As illustrated in the *t*-SNE plot, we were able to identify the original five NK clusters seen in the blood of the healthy donors with an additional cluster #6 (Fig. 9b). This new cluster was mainly comprised of cells from the donor with GATA2$^{T354M}$ mutation (Fig. 9c). Consistent with the flow-based identification of CD56$^{bright}$ NK cells, scRNA-seq analyses also revealed only 2% of the total NK cells were in the "CD56$^{bright}$ NK" cluster of the GATA2$^{T354M}$ donor (Fig. 9d). The composition of the other four clusters was relatively similar among the GATA2$^{T354M}$ donor and the two healthy controls (Fig. 9c, d). These results indicate that *GATA2* mutation indeed results in the loss of CD56$^{bright}$ NK cells instead of the loss of CD56 expression.

Next, we focused on the analyses of Cluster #6 due to its uniqueness to the *GATA2* mutation. The transcriptome signature of this cluster was substantially distinct to the previously identified five clusters as indicated by the Euclidean distance and module scores (Supplementary Fig. 10A, B). The GSEA results revealed significant enrichment of cell cycle gene sets in Cluster #6 compared to the rest of the cells (Fig. 9e). We calculated the cell cycle score and did not find substantially higher S.Score or G2M.Score of Cluster #6 (Supplementary Fig. 10C). The GSEA also found highly significant enrichment of transcription factors E2F and MYC in Cluster #6 compared to the rest of the cells, both involved in cell proliferation (Fig. 9e). From flow cytometry analyses, we did not find more cycling NK cells in the GATA2$^{T354M}$ donor compared to the control as indicated by Ki-67 staining (Supplementary Fig. 10D).

The majority of the GATA2$^{T354M}$ donor-derived NK cells clustered along with the healthy controls, indicating that the mutation so far has not caused a substantial transcriptome changes of the cells except for the Cluster #6. This could potentially result from the early stage of the disease in this donor. To compare the similarity between the GATA2$^{T354M}$ donor and the healthy controls, we used a Spearman's correlational matrix to compare the average gene expression among each cluster from each donor (Fig. 9f). It was evident that the "Active, Mature, and Terminal NK" clusters from GATA2$^{T354M}$ donor sample correlated well with the corresponding clusters in the healthy controls. On the contrary, the "CD56$^{bright}$ and Transitional NK" clusters did not correlate well between GATA2$^{T354M}$ donor and healthy controls. This result demonstrates that *GATA2* mutation renders considerable transcriptome alteration of CD56$^{bright}$ NK cells compared to CD56$^{dim}$ NK cells, consistent with more GATA2 protein in the CD56$^{bright}$ NK cells[17]. This could contribute to the specific loss of CD56$^{bright}$ NK cells.

Direct comparison between the GATA2$^{T453M}$ donor and the healthy controls revealed significantly increased expression of genes among the GIMAP family in the GATA2$^{T354M}$ donor sample. Within the top 10 most up-regulated genes in the GATA2$^{T453M}$ donor-derived NK cells, four of them belong to the GIMAP family (Fig. 9g, Supplementary Data 11). This family is known to play critical roles in the maintenance of the lymphocytes at steady state[64,65]. GIMAP4 has shown to be pro-apoptotic[66], whereas both GIMAP1 and GIMAP5 are anti-apoptotic[67–69]. These data imply an augmented pro-apoptotic propensity in the GATA2$^{T453M}$ donor NK cells. Indeed, flow analyses of freshly collected samples revealed more apoptotic NK cells in the GATA2$^{T453M}$ donor than the healthy control (Fig. 9h). Although the CD56$^{bright}$ NK cells are more viable than the CD56$^{dim}$ NK cells, the difference of the percentage apoptotic cells is larger in the CD56$^{bright}$ compartment than the CD56$^{dim}$ when comparing the GATA2$^{T354M}$ donor with the healthy control (Fig. 9h). While these data need further validation, it implies that the loss of CD56$^{bright}$ NK cells in patients with *GATA2* mutation could result from an increased apoptotic rate of CD56$^{bright}$ NK cells. Enrichment of gene sets related to mitochondrial functions stood out in the GSEA when comparing the GATA2$^{T354M}$ donor to the healthy controls (Supplementary Fig. 10E). However, with the staining of MitoTrackGreen and TMRE, we found cells from the GATA2$^{T354M}$ donor have lower mitochondrial mass and membrane potential compared to the healthy control (Supplementary Fig. 10F). The percentage of cells with polarized mitochondria defined by high expression of both mass and membrane potential was also less in the GATA2$^{T354M}$ donor (Supplementary Fig. 10G). Further exploration of metabolic changes in NK cells with *GATA* mutations is warranted.

Among the gene that was expressed at significantly lower levels in the GATA2$^{T453M}$ donor compared to the healthy controls, we found that the "Active NK" cluster-featured genes including IEGs are the genes that have highest difference in expression level

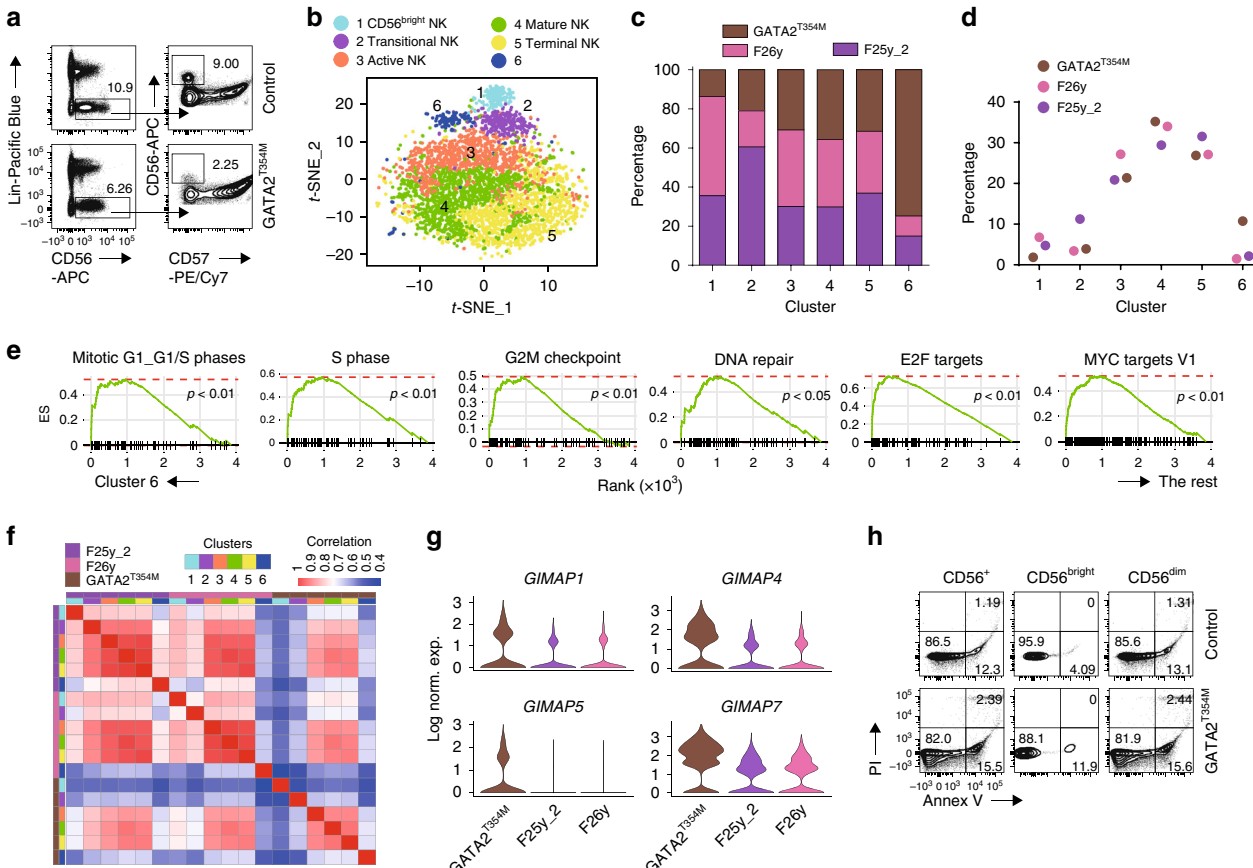

**Fig. 9** *GATA2* mutation results in larger transcriptome alteration in CD56^bright NK cells compared to CD56^dim NK cells. **a** The blood samples from the GATA^T354M donor and the healthy control were freshly collected and processed at the same time. Lin−CD56+ NK cells were evaluated via flow cytometry. The CD56^bright NK cells were identified with the help of CD57. **b** The combined analyses of the blood samples from the GATA2^T354M donor and the two healthy controls resulted in six clusters, as shown in the *t*-SNE plot. **c** Composition of the donors within each cluster. The input cell number from each donor is normalized to be equal. **d** Composition of the clusters within each donor. **e** Selected gene sets from the GSEA enriched in the Cluster #6 of the GATA^T354M donor sample compared to the rest of the cells. **f** Spearman's correlational matrix was used to compare the average gene expression among each cluster from each donor. **g** Four of the top 10 up-regulated DEGs of the GATA2^T354M donor NK cells belonged to the GIMAP family and were shown in violin plots. The *y*-axis represents log-normalized expression value. **h** The viability of NK cells from the GATA2^T354M donor and the healthy control were evaluated using Annexin V/PI staining. Source data for **c** and **d** are provided as a Source Data file. See also Supplementary Fig. 10

(Supplementary Fig. 10H, Supplementary Data 11). This indicates a potential defect of the homeostatic activation of NK cells in the GATA2^T453M donor. Along with this, the C3 gene set from GSEA revealed significant depletion of serum response factor (SRF) and ATF/CREB in the GATA2^T453M donor sample compared to the healthy controls (Supplementary Fig. 10I), key transcription factors involved in the expression of IEGs[36,37,70]. Maciejewski-Duval et al.[71] reported that surface *CXCR4* expression is diminished in NK cells of patients with *GATA2* mutation, but the total transcripts level is comparable to the healthy controls. However, in our dataset, we found nearly diminished *CXCR4* expression at the transcripts level compared to the healthy controls (Supplementary Fig. 10J). In summary, the reduced presentation of the features associated with "Active NK" cluster may further contribute to the developmental defects in patients with *GATA2* mutation.

## Discussion

We utilized scRNA-sequencing technology to explore the heterogeneity of human NK cells from both BM and blood of healthy donors and a donor with GATA2^T354M mutation. We uncovered significant heterogeneity than cell surface marker-defined NK subsets. A unique cluster with high expression of IEGs implies cellular activation of NK cells at steady state. The functionally

mature CD57+ NK cells are not a homogeneous population. Nearly half of them marked with high expression of *CX3CR1*, *HAVCR2* (TIM-3), and *ZEB2* demonstrate unique transcriptional features and may comprise the terminal mature NK cells. Moreover, we provide evidence at the transcriptome level in supporting the developmental progression from CD56^bright to CD56^dim NK cells and identify a transitional population. Finally, in a donor at an early disease stage caused by GATA2^T354M mutation, we confirmed the loss of CD56^bright NK cells via scRNA-seq. We found that this mutation resulted in larger transcriptome alteration of the CD56^bright NK cells compared to the CD56^dim NK cells, and the CD56^bright NK cells were dying faster than the healthy control. The reduced expression of IEGs also implies that the homeostatic activation is altered in patients with *GATA2* mutation, which potentially contributes to the NK cell defects.

The current human NK cell development model divides the Lin−CD56+ cells from the BM and blood into three distinct populations based on the surface expression level of CD56 and CD57 (CD56^bright → CD56^dimCD57− → CD56^dimCD57+)[6]. In our scRNA-seq analyses, we uncovered discrete developmental stages than these three simple subsets. We found a bona fide "CD56^bright NK" cluster in the scRNA-seq dataset, comprising

5–10% of total NK cells. We believe "Mature and Terminal" NK clusters together form the functionally mature CD57$^+$ NK cells. Although both *CX3CR1* and *HAVCR2* (TIM-3) marks the "Terminal NK" cluster at the RNA level, neither of them is sufficient to distinguish this population at cell surface level[55,56]. GSEA indicated dampened signaling and metabolic activity in the "Terminal NK" cluster compared to the "Mature NK" cluster. Whether there are differences in function and longevity between the "Mature" and "Terminal" NK cells remain an open question. We found a transitional population between CD56$^{bright}$ and CD56$^{dim}$ NK cells, supporting the developmental progression from CD56$^{bright}$ to CD56$^{dim}$ NK cells at the transcriptome level. Based on the expression of CD16 (*FCGR3A*), we reasoned that cells with intermediate cell surface expression of CD16 are potential cells in the "Transitional NK" cluster, existing in a small portion of CD56$^{bright}$ NK cells and part of the CD56$^{dim}$CD57$^-$ NK cells. Based on flow cytometry analyses, there are more NK cells with intermediate CD16 expression in the BM than the blood, consistent with a higher percentage of the "Transitional NK" cluster in the BM than the blood of the scRNA-seq data. The scRNA-seq data also picked up the adaptive NK cells marked with high expression of NKG2C (*KLRC2*). Interestingly, within the "Adaptive NK" cluster, only one donor had significant expression of *KLRC2*. This implies that adaptive NK cells arise from infections other than HCMV potentially share similar features with HCMV-induced NK cell memory.

One cluster that we found in both the BM and blood was the "Active NK" cluster. This cluster had high expression of IEGs, including *FOS*, *FOSB*, *JUN*, *JUNB*, and so on. More than 50% of the up-regulated DEGs of this cluster were transcription factors. We hypothesize that this could reflect homeostatic activation of NK cells via certain stimuli. Interestingly, Crinier et al.[60] have recently found a NK cell population in the mouse spleen with similar active transcriptional signatures, though they did not find this cluster in their human dataset potentially due to low cell number and sequencing depth or even the age and healthy conditions of the donors. This implies a conserved homeostatic activation of NK cells in both mice and humans, which is potentially critical to the survival, proliferation, or even differentiation of NK cells.

*GATA2* mutation has been of interest to the human NK cell biology due to the specific loss of CD56$^{bright}$ NK population[17]. The nearly complete loss of CD56$^{bright}$ NK cells at the time of the diagnosis of *GATA2* mutation impedes any study of the mechanism. The identification of a donor with GATA2$^{T453M}$ mutation at early disease stage with a reduced number of CD56$^{bright}$ NK cells provided us a rare opportunity to address this phenomenon. Using scRNA-seq, we confirm the loss of the CD56$^{bright}$ NK cells from this patient. Although the heterogeneity of the NK cells in this donor was mostly intact compared to healthy control, we found a more substantial transcriptome alteration in the "CD56$^{bright}$ and Transitional NK" clusters than the rest of clusters. The CD56$^{bright}$ NK cells also have a higher apoptotic rate than normal. Direct transcriptome comparison revealed significantly reduced expression of genes related to "Active NK" cluster in the GATA2$^{T453M}$ donor, implying impaired steady-state activation. This may also contribute to the NK defects in patients with *GATA2* mutation.

Unlike the murine model, there are more NK progenitors and immature NK cells in the secondary lymphoidal organs of the human. It is valuable to explore the heterogeneity of NK cells in those anatomic locations using this technology. Although Crinier et al.[60] used scRNA-seq to profile the human NK cells from the spleen, little information was uncovered related to the early developmental stages, presumably due to the sorting of only the

CD56$^+$ NK cells. Moreover, with the limit RNA being captured at the single-cell level, we do not have much success in determining the driving forces in promoting NK cell maturation, for example, the identification of unique transcription factors at distinct developmental stages. This problem will be solved with more advanced scRNA-seq technology in the future. Nevertheless, our study explored the heterogeneity of human NK cells at the transcriptome level and substantially expanded our understanding of human NK cells. For future application, the machine learning-based classifier algorithm could be built based on our datasets to characterize the developmental heterogeneity of human NK cells in patients with various genetic or pathological conditions.

## Methods

**Tissue collection and materials**. All healthy human BM and blood were de-identified samples. All fresh BM and blood samples from healthy donors for the scRNA-seq and flow cytometry experiments were obtained from the Stem Cell and Xenograft Core of the University of Pennsylvania, PA. All samples were provided anonymously after informed consent. Collection, distribution, and usage of these de-identified human materials were approved by the Institutional Review Board (IRB) of the University of Pennsylvania. The samples were shipped overnight and processed immediately upon receipt. Buffy coats from healthy donors used for flow cytometry experiments were ordered from Blood Center of Wisconsin, WI. All samples from the Blood Center of Wisconsin were provided anonymously after informed consent. Use of these human materials was approved by the IRB of the Blood Center of Wisconsin. The blood samples from the GATA2$^{T354M}$ donor and the corresponding healthy control were obtained from the Children's Hospital of Wisconsin under an approved IRB by the Medical College of Wisconsin. These two samples were obtained at the same time and processed immediately after collection. Recombinant Human Annexin A1 is from R & D (Minneapolis, MN). The corticosterone and human Annexin A1 ELISA Kits are from Cayman Chemical (Ann Arbor, MI).

**Cell separation cell sorting and flow cytometry**. BM and blood sample were diluted with ice-cold phosphate-buffered saline (PBS) containing 2 mM EDTA and carefully layered over lymphoprep (STEMCELL Technologies, Vancouver, Canada), and then centrifuged at 440 × *g* for 35 min at 20 °C without brake. After aspirating the upper layer, mononuclear cells at the interphase were carefully transferred and washed once with PBS containing 2 mM EDTA before downstream process. NK cells were sorted as CD3E/CD19/CD14$^-$CD7$^+$ cells using FACSAria III or FACSMelody (BD Biosciences, San Jose, CA), and the purity was generally above 95%. For functional assay, NK cells were isolated from peripheral blood mononuclear cells using negative selection kit (STEMCELL Technologies, Vancouver, Canada). Flow cytometry analyses were conducted in LSR-II (BD Biosciences, San Jose, CA) or MACSQuant Analyzer 10 (Miltenyi Biotec, Bergisch Gladbach, Germany) and analyzed with FlowJo software (FlowJo LLC, Ashland, OR). Lin refers to CD3E, CD19, CD14 or CD3E, CD19, CD14, CD20, and CD34. CD3E (UCHT1, 300417, @ 1:100), CD19 (HIB19, 302224, @ 1:100), CD14 (HCD14, 325616, @ 1:100), CD20 (2H7, 302320, @ 1:100), CD34 (581, 343512, @ 1:100), CD7 (CD7-6B7, 343108, @ 1:100), NKp80 (5D12, 346706, @ 1:50), CD16 (B73.1, 360708, @ 1:100), CD44 (IM7, 103024, @ 1:300), CXCR4-biotin (12G5, 306504, @ 1:50), CD57 (HNK-1, 359610, @ 1:50), TCRA/B (IP26, 306719, @1:100), TCRG/D (B1, 331221, @ 1:100), IFN-γ (4S.B3, 502512, @ 1:100), and streptavidin-PE/AF647 (405203/ 405207, @ 1:100) were from BioLegend (San Diego, CA); CD56 (TULY56, 11-0566-42, @ 1:100), CD69 (FN50, 12-0699-42, @ 1:50), CD62L (Dreg56, 12-0629-42, @ 1:50), Ki-67 (SolA15, 12-5698-82, @ 1:300), MitoTrack-Green, and TMRE are from Thermo Fisher Scientific (Waltham, MA); NKG2A (REA110, 130-114-089, @ 1:50), and NKG2C (REA205, 130-119-814, @ 1:50) are from Miltenyi Biotec (Bergisch Gladbach, Germany). Annexin A1 (D5V2T, 80994, @ 1:50) is from Cell Signaling (Danvers, MA); XCL1 (109001, MAB6951, @ 1:200) is from R & D (Minneapolis, MN). Annexin V Apoptosis Detection Kit is from BD Pharmingen (San Jose, CA).

**Single-cell RNA-sequencing**. Cells were washed once with ice-cold PBS containing 10% fetal bovine serum post sorting and counted using hemocytometer. After that, the cells were loaded to 10X chromium machine (10X Genomics, San Francisco, CA) and run through the library preparation procedures following guidance from the Chromium Single Cell 3′ Reagent Kits v2. The libraries were quantified using NEBNext Library Quant Kit (New England Biolabs, Ipswich, MA) and sequenced via Illumina NextSeq 550 (Illumina, San Diego, CA).

**Data analyses**. Following the sequencing, the raw data from each sample were demultiplexed, aligned to the hg19 reference genome, and UMI counts were quantified using the 10X Genomics Cell Ranger pipeline (v2.1.1, 10X Genomics). Then, we continued the data analysis with the filtered barcode matrix files using the

Seurat package (v2.3.1)[72] in R (v3.4.3 or above). For the initial QC step, we filtered out the cells that expressed <200 genes or >2500 genes. We also removed cells with >5% mitochondrial transcripts content. To avoid extra stimulation of the cells, we did not lyse the red blood cells (RBCs) before the scRNA-seq procedures. Thus, we detected several hemoglobin genes across the samples. This was unlikely due to conjugates formed by RBCs with NK cells as those should be filtered out from the 2500 genes cut-off. The released cytosolic contents from lysed RBCs during sample preparation potentially incorporated into the gel beads containing individual NK cells. As the expression of the hemoglobin genes was evenly distributed across all clusters, we reasoned that the hemoglobin genes or other transcripts from RBCs would not influence the clustering analysis stated below. Gene expression values for each cell were log normalized and scaled by a factor of 10,000. In order to prevent clusters from being biased by cellular library size or mitochondrial transcript content, gene expression values were scaled based on the number of UMIs in each cell and the cell mitochondrial transcript content. We combined cells derived from the same anatomic location of different donors to increase the power of unsupervised clustering analysis[73]. Due to the inherent variance among individuals, the cells formed a cluster based on their donor origin instead of different NK subsets. Therefore, we also scaled the donor IDs in addition to nUMI or mitochondrial transcript content. When we combined the NK cells from BM and blood of the same individuals for analysis, we only scaled the donor IDs but not the anatomic locations. Naive clustering of the cells into sub-populations was then conducted using Seurat's implementation of a shared nearest neighbor modularity optimization-based clustering algorithm (Louvain's original algorithm). Based on the PCElbowPlot, we picked a certain number of principal components (PCs) for the clustering analysis when that number reached to the baseline of the standard deviation of PC. We generally used the cluster resolution that higher one did not result in increasing clusters linearly. Cell clusters were visualized using t-SNE. For differential gene expression, we used model-based analysis of single-cell Transcriptomics (MAST) test[74] (log fc ≥0.25, min.pct = 0.1) and only selected the genes with adjusting $p$ value <0.05. For GSEA, we used fgsea function with gene sets from the Broad Institute's Molecular signatures database[75,76]. All the $p$ value shown in the figures were adjusted for multiple gene set enrichment comparison. In order to predict cellular differentiation, cells were ordered in pseudotime using Monocle2 (v2.6.4)[77]. The top 1000 DEGs across the five clusters (ranked by lowest $q$ value) were used to order the cells. The BEAM method within the Monocle2 package was used to identify genes that significantly differ in expression level across a branch point[77].

**Statistics**. The statistical significance was calculated using either paired Student's $t$ test or analysis of variance, as indicated in the figure legends. $P < 0.05$ was considered significant.

**Reporting summary**. Further information on research design is available in the Nature Research Reporting Summary linked to this article.

## Data availability

All scRNA-seq data that support the findings of this study have been deposited in NCBI GEO with the GSE130430 accession codes.

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

## Acknowledgements

We thank the help from Dr. Martin P. Carroll and the Stem Cell and Xenograft Core of the University of Pennsylvania. We also thank Mark Zogg for the guidance on the scRNA-seq experiment, Michael Reimer and Cynthia Opansky for help with the NextSeq 550 sequencer, and David Schauder and Kirthi Pulakanti for the teaching how to use scRNA-seq analyses software. This work was supported in part by NIH R01 AI102893 and NCI R01 CA179363 (S.M. and M.S.T.); HRHM Program of MACC Fund (S.M. and M.S.T.), Nicholas Family Foundation (S.M.); Gardetto Family (S.M.); and MACC Fund (M.S.T. and S.M.).

## Author contributions

C.Y. and S.M. designed and performed the research. C.Y., J.R.S., and R.B. analyzed the data. C.Y. wrote the manuscript. Z.J.G. conducted the mitochondria staining. A.R. conducted the exon sequencing of the GATA2^T453M donor. B.B. and S.R. provided technical support. M.R., K.-S.C., J.M.R., J.W.V., and M.S.T are the source of human samples. J.R.S., R.B., M.J.R., S.R., K.-S.C., M.S.T., and S.M. reviewed and edited the manuscript.

## Additional information

**Competing interests:** The authors declare no competing interests.

