## [Peer Review File · Nature Communications]

Reviewers' comments:

Reviewer #1 (NK cell biology, innate immunity)(Remarks to the Author):

In the present manuscript, Yang and collaborators investigated the human NK cell heterogeneity by a single-cell RNA sequencing (scRNA-seq) approach. By cell sorting, they purified subpopulations enriched in NK cells isolated from bone marrow or peripheral blood of healthy donors, that were subsequently used for scRNA-seq analyses. By tSNE the authors identified distinct cell clusters which are further dissected in detail along the manuscript. They used Differentially Expressed Genes (DEGs) identified in these subsets to infer about the potential biological significance of the clusters within the NK cell compartment. Finally, they performed an additional scRNA-seq on a patient bearing GATA2 T354M mutation in attempt to assess the potential consequences of this mutation on NK cell heterogeneity.

- The manuscript addresses the interesting research topic of NK cell heterogeneity. While the data reported are technically sound, the potential biological relevance of the findings remains unclear. Indeed, the work has no or little biological novelty. The main conclusion is that CD56dim NK cells derive from CD56bright cells. This message is hardly surprising and essentially confirmatory of data that are widely accepted. Furthermore, the authors speculate about the biological significance of different NK cell clusters basing on DEGs identified by scRNA-seq. This could be a valuable starting point, but the identification of cell subsets is not followed by functional experiments or description using converging methodologies or biology. This study would clearly require isolation of the NK cell clusters identified by the specific surface markers emerged from the transcriptomics profiles. Although this could be a challenging issue (at least for some NK clusters), this approach would allow to confirm and/or expand the properties of the cell subsets suggested by the transcriptomics.
- Information on BM NK cells do not allow to understand whether the authors are addressing nascent, resident or recirculating cells.
- Information is too much relying on small patient numbers and in some cases on single patient samples.

Reviewer #2 (Systems immunology, transcriptomic analysis)(Remarks to the Author):

In this report, the authors describe extensive scRNAseq profiling effort aiming at identification and characterization of human NK cell subsets. The depth of profiling is consistent with current practice in the field both in terms of the number of cells and reads. Analyses which have been conducted are also on par with standard practices. Profiling of NK cell populations in both blood and the bone marrow provide an original perspective. So does the inclusion of a donor with a genetic variation conferring a distinct cellular phenotype.

The overall quality of the work is good and it could be an important contribution to the field. Some points would, however, be worth addressing:

Comments:

- Publication should be contingent on the deposition of the data in a public repository such as the NCBI GEO. In addition, posting of the data on an interactive portal such as the one established at the Broad Institute could significantly raise the impact of this contribution.
- Claims that are made regarding originality of some of the findings should be revisited and investigated more carefully, for instance it is stated page 11: "The role of IL-32 in NK cells remain unknown" when evidence of a role for IL32, previously know as "Natural Killer Cell transcript 4", in NK cell immunobiology has been described in the literature (e.g. PMID: 22043900 (Park et al, 2012),

21321117 (Cheon et al, 2011), 15664165 (Kim et al, 2005).

- Other observations tend to be somewhat "soft", requiring for instance "further exploration with larger sample size" (p 17), for which meaning remains elusive (p 13), or where technical issues were present (p 16). This actually is not per se problematic, it is good practice to identify limitations of a study, but the authors may want to showcase one or two of the most compelling discoveries made and de-emphasize or even maybe omit some of the less solid observations.

- Most of the figures have too many panels, with some of the plots being as a result barely readable. Efforts should be made to show what is truly essential. Doing so definitely requires more effort but can only benefit the manuscript and the readers.

Reply to comments from Reviewers

(Comments from Reviewers – **black**; Responses – **blue**)

RE: NCOMMS-19-00350

"The Developmental Heterogeneity of Human Natural Killer Cells Defined by Single-cell Transcriptome"

Reviewer #1 (NK cell biology, innate immunity) (Remarks to the Author):

In the present manuscript, Yang and collaborators investigated the human NK cell heterogeneity by a single-cell RNA sequencing (scRNA-seq) approach. By cell sorting, they purified subpopulations enriched in NK cells isolated from bone marrow or peripheral blood of healthy donors, that were subsequently used for scRNA-seq analyses. By tSNE the authors identified distinct cell clusters which are further dissected in detail along the manuscript. They used Differentially Expressed Genes (DEGs) identified in these subsets to infer about the potential biological significance of the clusters within the NK cell compartment. Finally, they performed an additional scRNA-seq on a patient bearing GATA2 T354M mutation in attempt to assess the potential consequences of this mutation on NK cell heterogeneity.

- The manuscript addresses the interesting research topic of NK cell heterogeneity. While the data reported are technically sound, the potential biological relevance of the findings remains unclear. Indeed, the work has no or little biological novelty. The main conclusion is that CD56dim NK cells derive from CD56bright cells. This message is hardly surprising and essentially confirmatory of data that are widely accepted. Furthermore, the authors speculate about the biological significance of different NK cell clusters basing on DEGs identified by scRNA-seq. This could be a valuable starting point, but the identification of cell subsets is not followed by functional experiments or description using converging methodologies or biology. This study would clearly require isolation of the NK cell clusters identified by the specific surface markers emerged from the transcriptomics profiles. Although this could be a challenging issue (at least for some NK clusters), this approach would allow to confirm and/or expand the properties of the cell subsets suggested by the transcriptomics.

We thank the comments from the Reviewer. Specific to the biological novelty, we believe the “Active NK” and the “Terminal NK” clusters that we identified in the scRNA-seq dataset represent new information regarding the heterogeneity of NK cells. The “Active NK” cluster which has also been found in the spleen of mice (Crinier A. et al., *Immunity*, 2018 (PMID: 30413361)) represents an interesting population that is constitutively active at steady-state with high expression of immediate early genes (IEGs). The “Terminal NK” cluster is functionally-mature as that of the “Mature NK” cluster; yet harbors unique transcriptomic profiles indicating terminally-differentiated cells. Unfortunately, the discrepancy between the transcripts expression

and protein level impedes us from successfully identify these two populations by cell surface markers.

Specific to the “Active NK” cluster, we think it is not a unique developmental stage of NK cells but rather distinct NK cell subsets receiving similar stimulation at steady-state that results in this unique transcriptome profile. In this revised manuscript, we further explore this notion by evaluating the expression of IEGs in different subsets ($CD56^{\text{bright}}$, $CD56^{\text{dim}}CD57^{-}$, $CD56^{\text{dim}}CD57^{+}$) of human NK cells using the recently-published bulk RNA-seq dataset (Collins PL et al., *Cell*, 2018 (PMID: 30595449)). We found similar expression level of IEGs among these three subsets of NK cells, which further indicates that “Active NK” cluster may not represent a unique developmental stage. This data is included in the revised manuscript (**Figure S4D**).

Figure S4D

During this revision period, we focused on the identification of the “Terminal NK” cluster. The expression of *HAVCR2* (encoding TIM-3), *CX3CR1* and *ZEB2* is increased in this cluster compared to the rest. We decided to explore these three markers in identifying unique NK population. Consistent with previous work (Ndhlovu LC et al., *Blood*, 2012 (PMID: 22383801) & Hamann I et al., *Immunology*, 2011 (PMID: 21320123)), all the $CD56^{\text{dim}}$ NK cells express TIM-3 and *CX3CR1* as shown in **Figure A and B** below. Those two markers are not sufficient to distinguish specific populations. As for transcription factor *ZEB2*, we could detect minimal expression (**Figure C as below**). This is consistent with the fact that detection of *Zeb2* in murine model relies on the reporter system (van Helden MJ et al., *J Exp Med*, 2015 (PMID: 26503444)). Therefore, we could not use flow cytometry-based detection of *ZEB2* to distinguish any human NK subsets. These data are not shown in the revised manuscript.

Figure for reviewers only

As the reviewer suggested, the transcriptome information associated with each cluster represents important biological information that is worthy of exploration. We took the advice from the Reviewer and focused on the transcriptomic information related to the “CD56^{bright} NK” and “Mature NK” clusters as we could identify these two populations with cell surface markers.

The transcript levels of XCL1 and XCL2 were expressed at the highest in the “CD56^{bright} NK” cluster. XCL1 and XCL2 represent a unique class of chemokine that has been shown to be critical for dendritic cells (DCs) recruitment during infection (Lei Y and Takahama Y, *Microbes and Infection*, 2012 (PMID: 22100876)). Previous work from mass spectrometry has revealed higher protein level of XCL1 in the CD56^{bright} NK cells compared to the CD56^{dim} NK cells at steady state (Scheiter M et al., *Mol Cell Proteomics*, 2013 (PMID: 23315794)). However, the activation-induced production of this chemokine has not been explored in NK cells.

Through intracellular staining, we indeed found higher protein level of XCL1 in CD56^{bright} NK cells compared to the CD56^{dim} NK cells (**Figure 3C**). More importantly, when we stimulate freshly-isolated NK cells from PBMCs with PMA and Ionomycin, we found a robust induction of XCL1 expression in both CD56^{bright} and CD56^{dim} NK cells (**Figure 3C**). However, the production of XCL1 in the CD56^{bright} compartment is higher compared to the CD56^{dim} NK cells (**Figure 3C**). On the contrary, IL-12 and IL-18-mediated stimulation only induced the production of IFN- γ but not XCL1 (**Figure 3C and S3C**). Recent work demonstrated that NK cell-derived XCL1 and XCL2 are critical for recruiting dendritic cells into the tumor environment (Böttcher JP et al., *Cell*, 2018 (PMID: 29429633)). Given the prevalence of CD56^{bright} NK cells in the

secondary lymphoid organs, XCL1 and XCL2 production from CD56^{bright} NK cells may be important in both anti-pathogen and anti-tumor immunity.

Figure 3C

Figure S3C

In the “Mature NK” cluster, we found increased expression of genes belongs to the Annexin family (**Figure 6B**). Not much is known about the Annexin family proteins in the biology of human NK cells. Therefore, we decided to focus on Annexin A1 (*ANXA1*) as this protein has been shown to have immunosuppressive role in other immune cells and critical for the resolution of inflammation downstream of glucocorticoids (Perretti M and D’Acquisto F, *Nat Rev Immunol*, 2009 (PMID: 19104500)). NK cells are known to express Annexin A1 (Morand EF et al., *Clin Immunol Immunopathol*, 1995 (PMID: 7614738)). However, the relative protein level among different NK subsets are unknown. At the protein level, we could detect Annexin A1 intracellularly but not on the cell surface (**Figure 6C & S6A**). Consistent with the transcripts data, we found increased Annexin A1 protein as NK cells mature from CD56^{bright} → CD56^{dim}CD57⁻ → CD56^{dim}CD57⁺ population (**Figure 6C**).

Next, we tried to explore the role of glucocorticoids–Annexin A1 axis in the effector functions of human NK cells. Consistent with previous data (Morgan DJ and Davis DM, *Front Immunol*, 2017 (PMID: 28450865)), corticosterone inhibits the IFN-γ generation after activation (**Figure S6B**). However, Annexin A1 does not affect either PMA and Ionomycin- or IL-12 and IL-18-mediated IFN-γ generation (**Figure S6C**). We reasoned that this could due to low expression of Annexin A1 receptor, formyl peptide receptor 2 (*FPR2*) on NK cells (**Figure S6D**). Nevertheless, the storage of Annexin A1 in the cytosol of human NK cells potentially regulates the activities of other innate and adaptive immune cells through paracrine activation. To test this possibility, we explored the activation-mediated secretion of Annexin A1 from human NK cells. We found reduced protein levels of Annexin A1 in NK cells following stimulation with either IL-12 and IL-18 or PMA and Ionomycin indicating potential secretion of this protein (**Figure 6D**). To further test this hypothesis, we used ELISA to measure the protein levels of Annexin A1 in the culture supernatants following stimulation. Indeed, we found increased level of Annexin A1 in the culture supernatants after stimulation (**Figure 6E**). These data indicated that, in addition to neutrophils, monocytes, and macrophages, NK cells are also a major producer of

Annexin A1. Specific activation induces the secretion of Annexin A1 from NK cells, which could influence the inflammatory response.

Figure 6

Figure S6

Additionally, in the previously-submitted manuscript, we used the module score function based on differentially-expressed genes (DEGs) to evaluate the similarity among clusters and simulated the transcriptome progression (**Figure 3**). Based on this information, we identified the “Transitional NK” cluster. With the availability of the bulk RNA-seq dataset of the three established NK subsets (Collins PL et al., *Cell*, 2018 (PMID: 30595449)), we decided to utilize the DEGs from CD56^{bright} and CD56^{dim}CD57⁺ NK cells and evaluate the expression level of those genes in all the NK clusters defined in our scRNA-seq dataset through module score function. As shown in **Figure 3F and S3F**, we found the “Transitional NK” cluster has intermediate expression of signature genes of either CD56^{bright} or CD56^{dim}CD57⁺ NK cells. This data further confirmed the identity of the “Transitional NK” cluster.

Figure 3F

Figure S3F

- Information on BM NK cells do not allow to understand whether the authors are addressing nascent, resident or recirculating cells.

We thank the comment from the Reviewer. We agree that our BM population could contain nascent, resident, or recirculating cells. However, with human samples, it is extremely difficult to distinguish these three NK populations.

- Information is too much relying on small patient numbers and in some cases on single patient samples.

Most of the patients with GATA2 mutation exhibit a severe immunodeficiency with the absence of CD56^{bright} NK cells at the time of diagnosis. We were fortunate to identify this 33-year old female donor who was in the onset of the disease who possessed reduced yet detectable number of CD56^{bright} NK cells. This enabled us to explore the molecular mechanisms that underline the unique loss of CD56^{bright} NK cells in patients with GATA2 mutation. Our findings provide the first mechanistic explanation for the preferential loss of CD56^{bright} NK cells caused by the GATA2 mutation. Admittedly, we could not draw a definitive conclusion from a single patient sample. However, we strongly believe that our results implications for future studies related to the role of GATA2 mutations in NK cell development. However, if the Reviewer and/or the Editor feel strongly against this single patient data, we are willing to remove this dataset from the current version of the manuscript.

Reviewer #2 (Systems immunology, transcriptomic analysis) (Remarks to the Author):

In this report, the authors describe extensive scRNAseq profiling effort aiming at identification and characterization of human NK cell subsets. The depth of profiling is consistent with current practice in the field both in terms of the number of cells and reads. Analyses which have been conducted are also on par with standard practices. Profiling of NK cell populations in both blood and the bone marrow provide an original perspective. So does the inclusion of a donor with a genetic variation conferring a distinct cellular phenotype.

The overall quality of the work is good and it could be an important contribution to the field. Some points would, however, be worth addressing:

Comments:

- Publication should be contingent on the deposition of the data in a public repository such as the NCBI GEO. In addition, posting of the data on an interactive portal such as the one established at the Broad Institute could significantly raise the impact of this contribution.

We appreciate the comments and suggestions from the Reviewer. We have uploaded the entire dataset into NCBI GEO. The GEO accession number is GSE130430. Our data will be available to the public upon manuscript acceptance.

- Claims that are made regarding originality of some of the findings should be revisited and investigated more carefully, for instance it is stated page 11: “The role of IL-32 in NK cells remain unknown” when evidence of a role for IL32, previously know as “Natural Killer Cell transcript 4”, in NK cell immunobiology has been described in the literature (e.g. PMID: 22043900 (Park et al, 2012), 21321117 (Cheon et al, 2011), 15664165 (Kim et al, 2005).

We thank the comment from the Reviewer. We have corrected this statement and included the work that previously investigated the role of IL-32 in NK cell biology in the revised manuscript.

- Other observations tend to be somewhat “soft”, requiring for instance “further exploration with larger sample size” (p 17), for which meaning remains elusive (p 13), or where technical issues were present (p 16). This actually is not per se problematic, it is good practice to identify limitations of a study, but the authors may want to showcase one or two of the most compelling discoveries made and de-emphasize or even maybe omit some of the less solid observations.

We thank the comment from the Reviewer. We have revised these three statements to either omit some of the limitation or de-emphasize them.

- Most of the figures have too many panels, with some of the plots being as a result barely readable. Efforts should be made to show what is truly essential. Doing so definitely requires more effort but can only benefit the manuscript and the readers.

We thank the Reviewer's comment. We have made significant efforts to revise the plots so that they could be more reader-friendly. For all the VlnPlots, we have removed the dots so that it is easier for the readers to compare the value across different clusters. Specific to the heatmaps in **Figure 4** and **7**, we decided to demonstrate the average expression level of the genes among a cluster instead of individual cells to increase the readability. All the changes are reflected in **Figure 1C, 2C, 3A, 4B, 4D, 4E, 5A, 5D, 5E, 6A, 6B, 7A, 7B, 9G, S2A, S2C, S4A, S5C, S5D, S5F, S5G, S6E, S6F, and S9J.**

REVIEWERS' COMMENTS:

Reviewer #1 (Remarks to the Author):

The authors attempted to address some of the major concerns by exploring recent publicly available RNA-seq data to support their model and performed some functional experiments on NK cells focused on selected DEGs derived from their scRNA-seq. The analysis of the external RNA-seq dataset partially justify some of the authors' conclusions. Data on XCL1, XCL2 and Annexin-1 suggest a potential usefulness of their scRNA-seq data to infer novel insights on NK cell biology. However, the authors were unable to identify cell markers suitable for isolation of some cell clusters and to provide the necessary information on the properties of these cell subsets. Thus, the biological and/or clinical significance of the clusters identified remains undefined. In addition, their conclusions regarding a single patient bearing the GATA2 T354M mutation are essentially speculative.

Reviewer #2 (Remarks to the Author):

All comments have been addressed satisfactorily by the authors.